



# On the combined use of rain gauges and GPM IMERG satellite rainfall products: testing cellular automata-based interpolation methodology on the Tanaro river basin in Italy.

Annalina Lombardi[1,2,X], Barbara Tomassetti[1,X], Valentina Colaiuda[1,X], Ludovico Di Antonio[3], Paolo Tuccella[1,2], Mario Montopoli[6], Giovanni Ravazzani[5], Frank Silvio Marzano[1,4,†], Raffaele Lidori[1], Giulia Panegrossi[6]

[1]CETEMPS, Centre of Excellence University of L'Aquila, via Vetoio, 67100 Coppito (L'Aquila), Italy
[2]Department of Physical and Chemical Sciences, University of L'Aquila, via Vetoio, 67100 Coppito (L'Aquila) Italy
[3]Univ Paris Est Creteil and Université Paris Cité, CNRS, LISA, F–94010 Créteil, France
[4] Department of Information Engineering, Electronics and Telecommunications, Sapienza University of Rome, 00184 Rome, Italy
[5] Department of Civil and Environmental Engineering, Politecnico di Milano
[6] Institute of Atmospheric Sciences and Climate (ISAC), National Research Council (CNR) 00133, Rome, Italy

[X] These authors contributed equally to this work.

*Correspondence to*: Annalina Lombardi (annalina.lombardi@aquila.infn.it) and Barbara Tomassetti (barbara.tomassetti@aquila.infn.it)

**Abstract.**

The uncertainty of hydrological forecasts is strongly related to the uncertainty of the rainfall field due to the nonlinear relationship between the spatio-temporal pattern of rainfall and runoff. Rain gauges are typically considered as reference data to rebuild precipitation fields. However, due to the density and the distribution variability of the raingauge network, the rebuilding of the precipitation field can be affected by severe errors which compromise the hydrological simulation output. On the other hand, retrievals obtained from remote sensing observations provide spatially resolved precipitation fields improving their representativeness. In this regard, this paper aims to investigate the impact of using the merged rainfall fields from the rain gauge Italian rainfall network and the NASA Global Precipitation Measurement (GPM) IMERG precipitation product on the hydrological simulation performance. In particular, one aspect is to highlight the benefits of applying the Cellular Automata algorithm to pre-process input data in order to merge them and reconstruct an improved version of the precipitation field.

The cellular automata approach is evaluated in the Tanaro River Basin, one of the tributaries of the Po River in Italy. As this site is characterized by the coexistence of a variety of natural morphologies, from mountain to alluvial environments,



as well as the presence of significant civil and industrial settlements, it makes it a suitable case study to apply the proposed approach. The latter has been applied over three different flood events occurred from November to December 2014. The results confirm that the use of merged gauge-satellite data using the Cellular Automata algorithm improves the

performance of the hydrological simulation, as also confirmed by the statistical analysis performed for seventeen selected quality scores.



## 1 Introduction


Hydrological models are important tools for flood early warning system and management of water resources under climate change conditions. The accurate estimation of precipitation and its spatial variability within a watershed is crucial for reliable discharge simulations: the relationship between the distribution of precipitation and the calculated flow discharge is not linear; therefore, the precipitation patterns strongly influence the calculation of the runoff (e.g., Goodrich

et al.; 1997, Singh, 1997; Cristiano et al, 2017).

As far as the operational activity is concerned, the hydrological models are usually forced both with observed and predicted rainfall data, and the uncertainty of hydrological forecasts is strongly related to the uncertainty of the input rain field. Therefore, forcing the hydrological models with observed precipitation data as realistic as possible is essential to reduce their uncertainty.


The rain gauge data are typically used as the main source of information (Nikolopoulos et al., 2010) to produce an Areal Precipitation Estimate (hereafter APE), even if the reproduced rainfall spatial pattern can be affected by several errors. Furthermore, rain gauges, being in-situ captative-type instruments, can be considered highly accurate only over a limited area surrounding the instrument itself. Consequently, they have a reduced capability to represent the spatial distribution in highly variable precipitation fields, such as over complex terrain which are typically poorly gauged and

where the orographic precipitation effects take place. Increasing the density of the network can be a way to improve the representativeness of precipitation derived from rain gauges. WMO has established standard rules in terms of the minimum density needed to build precipitation measurement networks (Sevruk, 1992; WMO, 1994; Liang et al., 2012). However, such a standard cannot be always strictly followed due to practical reasons (e.g. geomorphological characteristics, environmental conditions, and the micro-climatic variability of the considered region). Accordingly,

several regional, national, and private rain gauge networks are generally not sufficiently distributed to fully satisfy the hydrological needs.

Nevertheless, the rain gauges still represent the main source of information to spatialize precipitation. The spatialization process considers the horizontal correlation structure of rain, leading to the definition of a correlation length or radius of influence through which rain gauge measurements are extended over unobserved (i.e., ungauged) surrounding

areas (e. g. Duque-Gardeazábal et al., 2018). However, the raingauge radius of influence may depends on location, time, event type (eg. convective or stratiform), network density (Gandin 1970), as well as the interpolation method implemented (Xu et al., 2013; Chacon-Hurtado et al., 2017; Andiego et al., 2018), thus leading to large uncertainties in the final APE and consequently in a reduced ability to model hydrological processes.

Remote sensing observations can represent a valuable gap filling tool, complementing the above-mentioned

limitations related to the APE. In particular, since satellite observations are spatially resolved, it opens to the more direct



use of Satellite-based RainFall Estimation (hereafter SRFE) (Li et al, 2021) into hydrological models. In this work, the role of APE from rain gauges and SRFE in the hydrological models is investigated. Indeed, it is well recognized that the accuracy of the results of many hydrological calculations depends on those of APE (see Nemec, 1986). The usage of SFRE for hydrological applications depends upon the type of application, the accuracy, spatial and temporal resolution

as well as the latency of the estimates: different applications have different data requirements. Kidd and Levizzani (2011) demonstrated that hydrological requirements for precipitation estimates can be divided into two main categories: high and lower resolution estimates for short- and longer-lived events, respectively. Flash flood events with rapid catchment response, necessitate of a fine spatial and temporal resolution, together with timely delivery of those estimates. Fluvial flooding and water resources are characterized by relatively long lead times and therefore some requirements can be

relaxed. As a matter of fact, it has been shown that SFRE's measurement uncertainties are associated to the intensity, the duration, and the scale of the event, showing an uncertainty decrease during higher rain rates, larger domains, and longer integration time: the more the precipitation tends toward deep convection regime, the more accurate the satellite estimates are (Maggioni and Massari, 2018; Maggioni and Massari, 2019). High-mountain regions are among the most challenging environments for remote-sensing-based precipitation measurements due to extreme topography and large weather and

climate variability. These regions are typically characterized by a lack of in situ measurements and hit by devastating flash floods (Dinku et al., 2007; Hong et al., 2007; Kubota et al., 2009; Tian and Peters-Lidard, 2010; 2010; Hirpa et al., 2010; Yong et al., 2010; Ghulami et al., 2017; Guo et al., 2017). In this regard, satellite sensors provide global coverage and observations in regions where in situ data is unavailable or sparse. Because of this peculiarity, the use of satellite data for hydrological applications has gained an increased interest, also given the significant activity of space-based

precipitation estimation techniques in the past few decades (Guetter et al., 1996; Tsintikidis et al. 1999; Wilk et al., 2006; Hughes, 2006; Su et al., 2008; Collischonn et al.,2008; Thieming et al. 2013; Jiang and Wang, 2019; Darko et al. 2021). However, limitations associated with the use of satellite rainfall estimates for hydrological applications related mainly to the error structure of satellite rainfall estimates (McCollum et al. 2002; Gebremichael and Krajewski 2004; Hossain and Anagnostou, 2006; Ebert et al. 2007; Dinku et al. 2007; Kirstetter et al. 2013; Maggioni et al. 2011, 2014, Falck et al.

2021) and to the rainfall error propagation through the hydrological model (Nijssen and Lettenmaier 2004; Hossain and Anagnostou 2005; Hong et al. 2006; Mei et al., 2017; Solakian et al. 2020; Camici et al., 2020; Brocca et al., 2020; Camici et al., 2022; Tramblay et al. 2023) should be considered.

The error propagation of satellite rainfall through hydrological simulation is related to many factors, such as specifications of the satellite rainfall product, basin size, spatial and temporal hydrological resolution, the used hydrologic model, and

geomorphological characteristics of the area (Mei et al. 2022). Dembélé et al. (2020) highlighted that although satellite products are characterized by uncertainties, their most reliable key feature is the spatial patterns representation, which is a unique and relevant source of information for distributed hydrological models. Their results demonstrate that there are



benefits in using satellite data sets when suitably integrated in a robust model parametrization scheme. Data integration was also recognized by Shi et al. (2020) to be a key point. The work suggests that hydrological simulation results using

an appropriate method for precipitation merging data can provide valuable spatially distributed rainfall leading to a more rational flood flow simulation.

Several techniques to merge different data sets and reduce uncertainties in rainfall estimation are available based on physical approaches or statistical algorithms (e.g., French & Krajewski, 1994, Todini, 2001, Li and Shao, 2010). Blending of precipitation data from different sources involves a deep understanding of the source of the observations, their

characteristics, and their limits.

It should be noted that there are not many studies in scientific literature that provide information on hydrological processes, such as floods and flash floods, especially on small-scale basins and complex terrain.

This paper aims to achieve two main objectives: 1) validate Cellular Automata (hereafter CA) algorithm (Packard & Wolfram (1985)) to obtain a satisfactory synthesis of rain gauge data and a satellite rainfall product, focusing on small-

medium scale river basins; 2) assess the possible benefits in combining the rain gauges and SFRE to overcome the limitations provided by in situ measurements alone.

The basin studied in this work is characterized by a uniformly distributed altimetry profile, with about 27% of mountain area, allowing valuable testing of satellite data. The secondary goal is also to assess if using the combination of rain gauges and SFRE could overcome the limitations provided by in situ measurements alone.

The used data source is hourly rain gauge, obtained from 352 rain gauge stations into the selected domain, distributed by the Dewetra Platform (Italian Civil Protection Department and CIMA Research Foundation, 2014) and the Global Precipitation Measurement (GPM) Integrated Multi-satellite Retrievals for GPM (IMERG), half-hourly 0.1°x0.1° (roughly 10 km x10 km).

These data sources are used to generate different rainfall datasets, with mutual correction of their implicit error

characteristics. To merge the data into a single rain field, CA algorithm (Packard & Wolfram (1985)) has been implemented in the CETEMPS Hydrological Model (hereafter CHyM) (Coppola et al., 2007, Verdecchia et al., 2008b) and it is then used to test hydrological response to different input rain fields. Finally, the error evaluation is dealt with scoring metrics in terms of comparison between simulated and observed flow discharge.

The paper is organized as follows: the geographical framework of the study area is described in section 2; in section

3 a detailed description of the field data collection is presented whereas methods are presented in section 4. Then, in section 5 the application of the proposed approach applied to three different case studies is discussed and conclusions are drawn in section 6.



## 2 Study area

In the Piedmont region, the northwestern part of Italy, the Tanaro River is among the main right tributaries of

the Po River in terms of catchment length (276 km) and drainage basin size (8.324 km2), and with an average flow discharge of 123 m3/s. The river flows eastward across northern Italy starting in proximity of the France border, Monte Saccarello (2201 m) in the Ligurian Alps (Figure 1).

According to Degiorgis et al. (2013) the river is characterized by morphological variability. Three main areas associated with very different characteristics were defined: 1) the mountain zone, with a mean slope of about 6%, deep

riverbeds, and very steep catchments; 2) the mild zone, with a mean 1% slope, mildly steep catchments, and shallower riverbeds; 3) the alluvial zone, with very small values of slope.

The Tanaro is the only river among the right-bank tributaries of the Po, and it has an alpine origin, although the low elevation of the Ligurian Alps and their proximity to the sea do not allow for the formation of snowpack or glaciers large enough to provide a constant source of water during the summer; moreover, the Alpine zone constitutes only part

of the basin drained by the Tanaro River. For this reason, the flow discharge is subject to great seasonal variations with a regime more typical of an Apennine stream and a maximum flow discharge that can reach 1700 $m^3/s$, in spring and autumn, and a very low flow rate in summer. The natural flow discharge of the Tanaro river is strongly affected by the anthropic impact due to the fragmentation of the river channels, with dams and water regulation causing diversions between basins and irrigation. Some artificial sections intersect natural branches and some of these sub-basins are used

for hydropower generation. The artificial basins along the river and its tributaries are also used for flood control.

The river is exposed to severe events: it has been affected by at least 136 floods in 200 years (from 1801 to 2001). The most significant of these events occurred in November 1994, when the entire river valley was damaged (Marchi et al., 1996; Luino, 2002) and the sensor at Montecastello, located at the outlet of the river recorded a maximum flow discharge peak of 4350 $m^3/s$ (Autorità di Bacino del Fiume Po).

## 3 Observed Data


Precipitation data are recorded for the 2014 period on an hourly basis. The precipitation datasets are discussed below and include gauge dataset, satellite-only dataset and the flow discharge data on selected point stations distributed along the Tanaro river basin.





### 3.1 Rain Gauge data.

It is common to attribute an area of influence on a network of rain gauges: in detail the gauge is in the center of its circular area of influence, defined as the radius of influence, R. Shi et al. (2020) suggest that the radius of influence, also considered the average distance between stations, can be computed as:

$$R = \sqrt{\frac{S}{N}} \qquad (1)$$


where S is the area of the smallest circle which can cover all the rain gauges and the considered basin whereas N is the number of rain gauges considered. Reasonable station coverage means that the average radius associated with the rain gauge network should be at least comparable to the value associated with the rain bandwidth (Duque-Gardeazábal et al., 2018). In this study, S is the area of the Tanaro basin and N is the number of rain gauges in the basin (73 in the basin):

the average distance of the next station is about 11 km, but the stations are not distributed regularly. As will be discussed later, different values of *R* are selected for the different hydrological simulations. As discussed in Sec. 2, since the Tanaro basin is divided into three territorial sectors, the average rain gauge distance is computed for each of them (see Table 1).

### 3.2 Satellite-based rainfall estimates

The satellite precipitation product used in this study is the Global Precipitation Measurement (GPM) Integrated

Multi-satellite Retrievals for GPM (IMERG). The products provide quasi-global (60° N–60° S) precipitation estimates combining measurements from passive microwave (PMW) radiometers comprising the GPM Low Erath Orbit (LEO) satellite constellation and infrared (IR) geostationary (GEO) sensors. The IMERG product is also available in the form of post-real-time research data, i.e., IMERG Final, after monthly rain gauge analysis is received and considered. In this study, IMERG version 5 Final (IMERG-F) Uncalibrated (UNCAL) and Calibrated (CAL), half-hourly 0.1°x0.1° (roughly

10 km x10 km) (Sungmin et al., 2017) rainfall rate estimates have been used.

### 3.3 Observed Flow Discharge data.

Flow Discharge data ($m^3$ $s^{-1}$) are used to evaluate the hydrological model output in response to different precipitation inputs. However, several issues must be considered when the evaluation of deterministic hydrological models is used, including the need to validate them with very long observed flow discharge data time series. These data

are not always available, especially on small seasonal streams that are usually instrumented. In addition, estimates of river discharge data are associated with significant uncertainties due to various conditions such as rating curve





interpolation, extrapolation, unsteady flow condition, and seasonal variations in river roughness (Di Baldassarre and Montanari, 2009; Di Baldassarre and Claps, 2011).

Eight stations with long time series of flow discharge, available for the year 2014, are selected for this study. The stations are distributed over the basin, as shown in figure 1 (blue numbers) and they are representative of the different sub-basins contained in the Tanaro river basin.

## 4 Methodology

The methodology rationale is shown in Figure 2. It includes four main tasks: precipitation data gridding, precipitation data assimilation and merging, hydrological model simulation and error score calculation. Different combinations of precipitation are tested as input to the hydrological model and error scores are calculated accordingly in terms of flow discharge. The proposed technique for merging different measured rainfall at different spatial scales is based on the concepts of data assimilation (Bouttier and Courtier, 1999) with particular emphasis on the transformation of point data to areal data. Observed satellite and rain gauge data are gridded respecting the resolution set-up of the hydrological model: each value of rain data (satellite or gauge) is associated with a grid point *i-th* of coordinates (*l, m*) of the selected domain. Different rain scenarios are produced, using the original datasets or merged rainfall data; the hydrological model has been forced with the different rebuilt hourly rain fields to simulate flow discharges and to evaluate each scenario. Moreover, the model was not voluntarily calibrated ad hoc for this study, as it was used as a pure analysis tool.

### 4.1 Precipitation data gridding

The precipitation data gridding is necessary to speed-up the numerical processing in the hydrological model, and it defines, on a regular grid, a first guess in terms of precipitation field at the hydrological scale, hereafter termed as Precipitation Background Field (hereafter PBF) (Coppola et al, 2007). The Cressman algorithm (Cressman, 1959) is used to initialize the rain field grid points in the selected domain. Because of its simplicity, the Cressman method can be a useful starting point (Bouttier and Courtier, 1999). According to Li and Shao (2010) the used kernel function determines the accuracy of the fused field and to define a kernel function, it is also necessary to select rain radius. The radius of influence, $R$, determines the smoothness of the estimated field, containing the spread of the kernel function: a small $R$ corresponds to a rough estimated field and large variance, while a large $R$ corresponds to a smooth surface. Based on these considerations, given a discontinuous background field, the rainfall for each grid point of the selected domain is estimated as follows:

$$P_i = \sum_j \frac{1-\left(r_{ij}/R\right)^2}{1+\left(r_{ij}/R\right)^2} P_j \tag{2}$$





where $P_i$ is the estimated rain value at the *i-th* grid point, $P_j$ are the rainfall measurements available within the

radius of influence, *R*, and r$_{ij}$ is the distance between rain gauge location *j* and the grid point *i*.

Obviously, the first difficulty lies in selecting the reasonable value of *R*. Figure 3 shows the area coverage by

rain gauge network, when the algorithm uses a radius of influence equal to 5km. Even if observed data were available for

every grid point in the selected domain and no significant errors are found, the rainfall field rebuilt using a direct merging

method as the Cressman objective analysis scheme (e.g., Pereira Filho et al., 1998; Goudenhoofdt and Delobbe, 2008)

would produce significant bias at the boundary (Li and Shao, 2010; Duque-Gardeazábal et al., 2018); this means that a

smaller value of *R* would lead to a bias, but in a smaller area around the boundary. However, *R* selection is not a remedy

to the boundary bias because the rain bandwidth is likely to be large when observed points are distributed irregularly.

The problem of boundary bias is caused by the discontinuity of the background field due to the discretization of

the field, while the nonparametric merging method is only able to generate continuous surfaces. To overcome this issue,

a double smoothing merging method is applied. It is used to reduce the boundary bias (Li and Shao, 2010; Duque-

Gardeazábal et al., 2018) as better explained in the next section.

### 4.2 Precipitation data interpolation and merging: Cellular Automata technique

CA technique is used in this work as a double smoothing estimation. It is a simple mathematical idealization of

natural systems according to Packard & Wolfram (1985), based on the behavior that every single element of a natural

system can assume. CA can be described, for example, as identical discrete sites of a lattice, and the state of each grid

point evolves according to deterministic rules, conditioned by the values of neighboring cells at discrete time steps.

CA based algorithm has been developed and implemented in the hydrological model code. According to CA

theory, the input grid is considered an aggregate of cellular automata and the status of a grid point corresponds to the

value of a rebuilt (i.e. smoothed) precipitation field. The evolution of the precipitation status in the *i-th* grid point of the

lattice ($P_i^{(new)}$) is updated according to the following rule:

$$P_i^{(new)} = P_i + \alpha\left(\sum_{i=1}^{8} \beta_j (P_j - P_i)\right) \tag{3}$$

where $P_i^{(new)}$ is carried out over all 8 surrounding cells. The coefficients $\beta_j$ allow to consider the different

distances between the cells. As an example, for a regular equally spaced lattice, we assume the value 1 for the cells in

North, East, South and West location, and the value $1/\sqrt{2}$ for the cells located in the North-East, North-West, South-East

and South-West direction respect to the cell *i-th*. The coefficient $\alpha$ assumes a small value (typically from 0.1 to 0.9) to

ensure a slight smoothing of the original matrix: all grid points are updated synchronously, and the smoothing is performed





until the stability is reached, meaning that no significant changes are recorded in the calculated matrix. The grid point associated with the rainfall value available in the considered database is not modified by the algorithm. In terms of time evolution, a regular lattice is updated in discrete time steps according to the previous rule depending on the state of the site and the eight neighbouring cells. Therefore, the rule in Equation 3 can be written as follows:

$$P^{t-1} = P^t + \alpha \sum_{k=1}^{8} \frac{1}{r_k} P_k^t \qquad (4)$$

where $r_k$ is the distance between the considered cell and the neighbouring grid point. The value of rainfall in the cells is serially modified (eq. 4) and the sum is computed using only the neighbouring grid points. The CA method allows to perform the assimilation and spatialization of the rain field, it is useful for the high resolutions necessary for

hydrological simulations, and to use different sources of precipitation data. In this study, to test the assimilation of satellite rainfall data in the presence of sparse gauge stations, the CA algorithm has been implemented in the hydrological model, using two different assimilation approaches: NoModular and Modular.

In the NoModular approach, a high-resolution lattice is filled with both the satellite rainfall data, and the rain gauge data, used simultaneously at each time step, prioritizing the rain gauge data. To define a PBF, the approach uses an

R of 10 km, which corresponds to the satellite spatial resolution, the lowest resolution to cover the whole considered domain; CA technique is then applied. In the Modular approach, a hierarchical sequence of modules is used to assimilate the different data sets, making it possible to consider the different nature of the data. Therefore, the lattice of the considered domain can be divided into as many subdomains as the data sources type. Each subdomain can be defined as a set of grid points that have at least one rainfall value in a selected radius, R, whose value depends on the density of the

available data. Three different radii of influence were selected in this study to allow for different coverage of rain gauge data compared to the satellite data.

Using the CA technique, this study aims at identifying how different input data settings can affect the hydrological model performance, and if merging rain gauge and satellite rainfall data improves hydrological outputs. The degree of freedom of the input data settings are: 1) the type of data sources; 2) the data merging used approach: NoModular

or Modular; 3) the different values of the radius of influence, which determines the rain gauge data coverage area before the use of CA and 4) the satellite data used: Uncalibrated or Calibrated.



### 4.3 Hydrological modelling: CETEMPS Hydrological Model

The CHyM has been applied for climatological studies (Coppola et al., 2014, Sangelantoni et al., 2019), but it mainly has

been used as an operational tool for early warning systems (Tomassetti et al., 2005; Ferretti et al., 2019; Colaiuda et al., 2020; Lombardi et al., 2021). CHyM is a distributed, physically based hydrological model; hydrological processes (surface runoff, infiltration, evapotranspiration, percolation, melting and return flow) are explicitly simulated. In addition to being used to acquire different data sources or rebuild the spatial distribution of precipitation at hydrological model scale, the CA algorithm allows the model to simulate the hydrologic cycle of any defined geographic domain and at any

fixed spatial resolution up to the Digital Elevation Model (DEM) resolution (90 meters in the current version). The choice of spatial resolution is mainly related to the validity of the numerical schemes used to simulate hydrological processes (such as the shallow water kinematic wave used to solve the continuity equation which is considered a good approximation at a horizontal resolution of a few hundred meters). Using ChyM, the spatial domain is extended well beyond the investigated basin. This approach is useful to avoid boundary effects and to have a better rebuild of the precipitation field.

Furthermore, CHyM is a valid tool to investigate the rebuilt of the rainfall field, given that the resulting flow discharge value is linked exclusively to the rainfall, in fact the effects related to the base flow discharge are not visible, since the model does not reproduce them, given the short simulation spin-up time.

### 4.4 Error Score Metrics

To assess the fit between the observed and simulated flow discharge time series, objective functions were selected. The traditional performance indicators have been used, such as the Nash–Sutcliffe Efficiency (NSE) (Nash and Sutcliffe, 1970), and bias percentage (PBIAS) measuring the average tendency of the simulated values to be larger or smaller than the observed ones. The optimal value of PBIAS is 0.0, with low-magnitude values indicating accurate model simulations. Furthermore, the following scores were considered: Root Mean Square Error (RMSE), Mean Absolute

Relative Error (MARE), sensitive to extreme values (i.e., outliers) and to low values, Original Kling-Gupta Efficiency (KGE, Gupta et al., 2009), Modified Kling-Gupta Efficiency (KGEprime, Kingl eta al., 2012), Non-Parametric Kling-Gupta Efficiency (KGEnp, Pool et al., 2012).

According to Mathevet et al. (2006), KGE and NSE can be calculated in a bounded version: Bounded Nash-Sutcliffe Efficiency (NSEc2m), Bounded Original Kling-Gupta Efficiency (KGEc2m), Bounded Non-Parametric Kling-

Gupta Efficiency (KGEmp_c2m), Bounded Modified Kling-Gupta Efficiency (KGEprime_c2m). The analysis is carried out using an open-source evaluator for flow discharge time series (Hallouin, 2019). In addition to the conventional scores, other indicators were selected to obtain a more objective analysis, independent of the limits of the scores commonly used





for hydrological analyses, for a total of 17 quality scores. The idea is to consider the river flow discharge profile as a signal and for this reason, indicators, commonly used in generic signal studies, have been used.

The Match Correlation (MC) is the relationship between the Auto-correlation curve and the Cross-Correlation curve (Observed VS Simulated) and allows to understand the two curves overlap: the best value obtained will be close to 1.

$$MC = \frac{\int \text{AutoCorrelation\_of\_Observed\_values}}{\int \text{CrossCorrelation\_of\_Index\_values\_VS\_Observed\_values}} \tag{5}$$

The cross correlation (CC) is typically used in the signal theory for the assessment of similarity between two signals (Rabiner and Gold, 1975; Rabiner and Schafer, 1978; Benesty et al., 2004). The Correlation Time Delay (CT_D, Lombardi et al., 2021) represents an estimation of time shift between two series:

$$CT\_D = \max_{L \in R} CC(L) \tag{6}$$


the value of time lag $L$ maximizes the product obtained in the $CC$ calculation. Therefore, this quality score is suitable for measuring the effectiveness of the signal provided by hydrological simulations. The Time Peak Delay (TP_D) is a timing score and represents the hourly delay of the estimated maximum peak flow discharge compared to the observed one. The percentage Error (E%) at the peak value of the flow discharge was calculated as follows:


$$E = \frac{\max D_{Sim} - \max D_{Obs}}{\max D_{Obs}} \tag{7}$$

where $D_{sim}$ indicates the simulated flow discharge and $D_{Obs}$ represents the observed flow discharge.

The Dynamic Time Warping (DTW, Berndt and Clifford, 1994; Keogh and Ratanamahatana, 2005; Maier-
Gerber et al., 2019 and Di Muzio et al., 2019) find the similarity between two sequences by looking for the best alignment. For the N-by-M matrix, built using two discrete series $x(i)$ and $y(j)$ of $N$ and $M$ components respectively, a "warping" path $W$ is defined as a contiguous set of $L$ matrix elements, and the measure of misalignment $d$ for the path $W$ is given by:

$$d(W) = \frac{\sum_{i,j} V(i,j)}{\frac{1}{2}L(L-1)} \tag{8}$$






where the sum in the numerator is carried out over all the elements belonging to the warping path *W*. Each element *V(i,j)* represents the Euclidean distance between the i-*th* element of the first sequence and j-*th* element of the second sequence. The denominator is used to normalize different length sequences. The DTW index is then calculated as the minimum value of *d(W)*, considering all the possible path *W*.


$$DTW = \min_{W} \; d(W) \qquad (9)$$

The optimal path will be the N diagonal elements of matrix *V*, if the two considered sequences are aligned and have the same number of components (N=M). The DTW technique, however, could lead to wrong results in finding the optimal

alignment because a feature (e.g., a local peak or minimum) in one sequence is higher or lower than the corresponding feature in the other sequence. To overcome this issue, Keogh and Pazzani (2001) proposed the computation of warping using the local derivative of the time series to be compared: the Derivative Dynamic Time Warping (DDTW). The first derivative was calculated for each time series as follows:


$$D(x[i]) = \frac{(x[i]-x[i-1])+((x[i+1]-x[i-1])/2}{2} \qquad (10)$$

The main limitation linked to both analyses is defined singularities (Sakoe, & Chiba 1978; Keogh and Pazzani, 2001), i.e., the algorithm may try to explain variability in the Y-axis by warping the X-axis. This can lead to unintuitive alignments where a single point on one time series maps onto a large subsection of another time series. To overcome

these limits, we used the Windowing method (Berndt and Clifford, 1994). Allowable elements of the matrix can be restricted to those that fall into a warping window defined according to the following rule:

$$|i - (n/(m/j))| < \omega \qquad (11)$$

where *i* and *j* are the allowable points of the $n x m$ matrix, constrained to fall within a given warping window, $\omega$ a positive integer window width. In this work, $\omega$ is equal to 10 and this allows us to mitigate the effects linked to the baseflow discharge.



## 5 Analysis and discussion of the results

One of the effective strategies for the validation of satellite rainfall data is an indirect method through a hydrological assessment. It is worth noting that data on artificial water management are not available for the case study considered, therefore the hydrological model has difficulties in the presence of highly regulated basins since the simulation reproduces the natural river flow discharge without considering the human impact. Thus, a preliminary screening is carried out to minimize any anthropogenic impact in our analysis.


### 5.1 Analyzed case studies.

    In this study, the hydrological simulation ranges from November 1, 2014, to December 31, 2014. Since the purpose of this work is to investigate the performance according to the different rain scenarios (model forcing), November and December represent the most suitable period from the climatic point of view: in fact, according to the authors'
experience, the succession of rainfall events in the fall reduces the anthropic impact. In late fall-early winter dams and reservoirs are often at the limit of their capacity, allowing the water to laminate, the river flow discharge to be comparable to the natural one, which is simulated by the CHyM. Three time series, related to the three different flood events, have been studied:

- Case Study 01 - 10/11/2014 00UTC – 14/11/2014 23UTC
- Case Study 02 - 15/11/2014 00UTC – 20/11/2014 23UTC
- Case Study 03 - 29/11/2014 00UTC – 03/12/2014 23UTC

Figure 4a shows the synoptic charts of the fifth generation ECMWF reanalysis (ERA5) 500 hPa geopotential height and sea level pressure (Hersbach et al. 2023), related to the first analyzed case study (12 November 2014 00UTC). The European scenario was mainly characterized by the presence of a deep depression area located in the North Atlantic and
by a persistent blocking system of high-pressure on the Eastern continental sector. A trough associated to the oceanic depression was slowly moving toward Eastern Mediterranean by rotating its axis. This configuration caused instability conditions in Northern Italy, with widespread precipitation especially in the North-Western sectors, and cumulated rainfall up to 250 mm in 120 hours (10 November 00UTC – 14 November 23UTC) in the area of interest (Figure 4d).

    The synoptic scenario for the second case study (16 Novmber 2014 00UTC) resulted from a slow evolution of
that described above. As shown in Figure 4b, the circulation was slowed down by a high-pressure system located on Eastern Europe, extending from Anatoly up to North Sea, blocking the shift of the Oceanic trough toward the East. The most intense precipitation was registered in the Italian Northwestern sectors, with cumulated rainfall up to 250 mm in 120 hours (15 November 00UTC – 19 November 23UTC) in the area of interest (Figure 4e).





Figure 4c shows the synoptic situation related to the third case study (1 December 2014 00UTC). In this period,
the typical Western Mediterranean weather conditions were affected by the evolution of a deep cut-off low. On 29
November, it was centered on Morocco and in the following days, it moved eastward, advecting subtropical warm and
moist air towards northwestern Italy. The flux produced intense precipitations on the Ligurian territory, with cumulated
rainfall up to 150mm in 120 hours (29 November 00UTC – 03 December 23UTC) in the area of interest (Figure 4d).

Eight different hourly simulations have been carried out for each case study, using the eight different rain input
settings (Figure 2). Hourly hydrological simulations are possible thanks to the availability of the observed data: the
temporal resolution of the hydrological simulations, especially for small hydrological basins that have very short recharge
times, is very important, given that satellite data are provided every half hour, they are essential for developing operational
monitoring and forecasting tools for flood early warning systems. Therefore, UNCAL and CAL simulations use only
satellite data respectively IMERG-F Uncalibrated and IMERG-F Calibrated, GAUGE simulation uses rain gauge data;
the GAUGEUNCAL simulation uses the combined gauge and satellite data using the NoModular approach.
MODGAUGEUNCAL1, MODGAUGEUNCAL3, MODGAUGEUNCAL5 are the simulations where the hydrological
model has been forced using a Modular approach and different radii of influence related to rain gauge data merging gauge
and Uncalibrated satellite data (the number at the end of the simulation name is related to the gauge radius of influence:
1km, 3km and 5km); whereas the last hydrological simulation, MODGAUGECAL5 is carried out in the same way, but
using the Calibrated satellite data.

### 5.3 Results: hydrological simulation analysis

The experiment uses an indirect validation technique of precipitation data, through an analysis of the flow
discharge simulated by CHyM, where the model has been forced with eight rainfall scenarios. The APE produced using
IMERG F UnCalibatred (UNCAL) and Calibrated (CAL) and rain gauge data (GAUGE) separately, is based on different
$R$ for each dataset, as defined in Equation (1). Note that in the case of satellite data, $R$ is fixed to 10 km (the IMERGE
products resolution). In the case of the GAUGE, $R$ has been set at 30km, as in the CETEMPS hydrological operational
set-up (Colaiuda et al., 2020), to have a coverage of all the points of the grid in the considered domain. Therefore, even
if the number of rain gauges that fall in the analyzed basin, according to equation (1), gives an average distance of the
next station of about 11 km, in order to have a total coverage of the entire simulated domain (defined by the coordinates:
$43.9 \leq$ latitude $\leq 46.59$ and $6.49 \leq$ longitude $\leq 9.18$) and to account for the rain gauge spatial distribution, $R$ cannot be
lower than 30 km. In fact, to avoid boundary effects, the spatial domain has been extended well beyond the studied basin:
this strategy is useful for a better rebuilding of the precipitation field (Figure 1).



Figure 5 shows the CHyM rebuilt rain field for the Case Study 01. In detail, Fig. 5a represents 120h accumulated
rain carried out from GAUGE simulation obtained forcing the hydrological model with rain gauge data; Fig. 5b and Fig.
5c respectively represent CAL and UNCAL simulations, where CHyM has been forced using respectively GPM IMERGE
FINAL CAL and GPM IMERGE FINAL UNCAL. Figure 5f shows the 120h cumulated rain related to
MODGAUGEUNCAL5 simulation, obtained forcing the hydrological model with rain gauge data, using a Radius of
influence equal to 5km, merged with GPM IMERGE FINAL UNCAL.

The preliminary comparisons related to Case Study 01, between observed and simulated flow discharge data
with the different rainfall scenarios are shown: in Figure 6 Alba Tanaro and Ponte di Nava sections were selected for this
quick comparison. The hydrometric stations are in sections draining respectively 3385 km$^2$ and 145 km$^2$ upstream. From
a first intuitive analysis, the model appears to perform better with the GAUGE (cyan line), compared to the use of satellite
data only, both UNCAL (gray line) and CAL (yellow line) (Figure 6 a, c), consistently with the literature: rain gauges can
be considered as the most accurate approach for measurements. A sensitivity series of tests have been carried out and the
most significant are reported (figure 6 b, d). The tests refer to i) the used approach: NoModular or Modular; ii) the value
of $R$: 5 km, 3 km and 1 km for the rain gauge data and 10 km for satellite data respectively; iii) the satellite data used:
GPM IMERGE FINAL UNCAL and GPM IMERGE FINAL CAL.

From the analysis of Figure 6, it can be deduced that the best performances are obtained for MODGAUGECAL5,
although comparable performances are obtained using MODGAUGEUNCAL5 (indeed, given the small difference
between the two results, the two curves are graphically superimposed). In these cases, the background rain field of the
rain gauge data with $R = 5$ km (a coverage of 68% of the surface of the considered basin, Table 1), smoothed with CA, is
merged with the remaining part of the surface covered by the satellite data, where its background areal is first created
with a $R = 10$ km, filling the grid points left uncovered (32%) and then applying a definitive smoothing with CA. The
scores also confirm the best performances obtained by the simulation using the merged data. As an example, for the CS
01 at the ALBA section river, the KGE assumes values ranging from -1.16 (UNCAL), -0.95 (CAL), -0.703
(MODGAUGEUNCAL1), -0.366 (GAUGEUNCAL), 0.066 (GAUGE), passing to 0.209 (MODGAUGEUNCAL3), up
to 0.584 (MODGAUGEUNCAL5) and 0.585 (MODGAUGECAL5).

Figure 7 shows the boxplots for KGE and RMSE related to all Case Studies and all river sections and confirms
what has been shown so far: certainly, the rain gauge data allows for better performance than using the satellite data alone,
but the best results are obtained when the two data sources are merged, especially using the MODGAUGECAL5 setting,
which is comparable to MODGAUGEUNCAL5.

To obtain an objective evaluation, the statistical analysis has been performed using different quality scores,
evaluating their overall average (AVG) related to the three Case Studies and all stations (Table 2). Table 2 is divided into
two parts and the various settings have been placed following the order of increasing performance. In the first part of





Table 2 an improvement corresponds to increasing values, while in the second part an improvement corresponds to decreasing values. All scores confirm the results obtained from the comparison between observed and simulated flow discharges (Figure 6), showing better performance using the rain gauge data only (GAUGE) compared to satellite data only. Simultaneously, the calibrated satellite data (CAL) allows the model to perform better than the uncalibrated ones

(UNCAL). There is an evident improvement in the results obtained by merging the different sources of observed data (gauge and satellite) compared to simulations that use only satellite data.

The KGE score, for example, shows the above: an AVG ranging from -1.41 for UNCAL to 0.11 for GAUGE to 0.40 for MODGAUGECAL5 (Table 2). The MODGAUGEUNCAL3 has comparable performance, although lower, with respect to GAUGE (only rain gauge data) and to the MODGAUGEUNCAL5 and MODGAUGECAL5, where the rain

gauge data have a coverage of 68%. Although they do not improve compared to GAUGE, this is encouraging, suggesting that even with a lower rain gauge density the performance of the hydrological simulation can be guaranteed. MODGAUGEUNCAL1 has been used to test the minimum bandwidth of rain gauge size in the basin. Not all results are satisfactory; for example, in the case of NSE, a classic skill score in hydrology, it is a convenient and popular (albeit gross) indicator of the model's ability (there has been a long and lively discussion about its eligibility (Gupta et al., 2009)).

The simulation efficiency can be considered significant if the results are greater than 0: the study results do not respect these conditions, they are negative or at most close to zero. Although the results, our aim is to verify which APE obtained with the different settings improves the performance of the hydrological model and the results obtained with NSE test confirm it: the score goes from -15.618 for the UNCAL simulation (the worst performance) to -0.104 for MODGAUGEUNCAL5 and MODGAUGECAL5 simulations. Figure 8 shows the AVG values, listed in Table 2, of some

of the considered scores. In detail, Figure 8a shows some of the scores where the best performances are identified by a value equal to 1: KGEnp, NSEc2m, KGEc2m and MC.

In the second part of Table 2, the error is measured in terms of RMSE, MARE and PBIAS. In the case of RMSE and MARE, the trend of the results confirms what has been verified with the other quality scores. For what concerns PBIAS, the results are slightly different. The low magnitude of PBIAS indicates an accurate model simulation, positive

values indicate overestimation, whereas negative values indicate model underestimation. In this case, the best performances are evident for MODGAUGEUNACAL3, GAUGE, MODGAUGEUNCAL5 and MODGAUGECAL5 simulations, although their interpretation is not as straightforward as for other scores. In detail, the overall PBIAS AVG values are 8.22 for MODGAUGEUNACAL3, -13.33 for GAUGE, and about 15 for MODGAUGEUNCAL5 and MODGAUGECAL5 simulations. Also, in this case, as shown in Table 2 and Figure 8 b CT_D, TP_D, E%, DTW, DDTW,

an improvement in performance is confirmed with a clear decrease in the trend.





The comparison between the different Modular settings was necessary to verify if CA technique could overcome the limit of the satellite rain data calibration. In fact, using a 5km rain gauge radius of influence, the results are comparable both for calibrated and uncalibrated satellite data.

Regardless of the setting of the different runs, an improvement in results is obtained by merging rain gauges and
uncalibrated satellite data compared to using only calibrated GPM IMERG; an improvement was also found by merging the data compared with using only rain gauge data. Thus, the Modular approach with $R = 1$ km (MODGAUGEUNCAL1) provides a lower-performing hydrological simulation than the NoModular approach (GAUNGEUNCAL), while a strong performance improvement is evident with the Modular approach, where rain gauges have $R = 5$ km (MODGAUGEUNCAL5 and MODGAUGECAL5), compared with the simulation using only rain gauge data (GAUGE).
All the other scores reflect the expected trend: the performance of the model improves by merging the data, but above all, using this approach, whether calibrated or uncalibrated satellite data are used, the performances are comparable. In addition to using the average values (AVG) of the scores, the overall median (MED) and the overall standard deviation (STD) for the different runs have been computed and they are reported in the Supplementary materials (Table 3 and Table 4). A general increase in performance using merged data is obtained also for MED and for STD.

## 6 Conclusions

Hydrological models are important tools for flood early warning systems and management of water resources, especially under climate change conditions. The accuracy of the results of many hydrological computations depends on the accuracy of the Areal Precipitation Estimation. In fact, a more realistic rainfall distribution is as important as the correct estimate of the rain maximum cumulated values, especially when severe weather events affect areas with a
complex drainage network and characterized by small-medium size river basins, very close to each other. Accurate estimation of APE based on the measurement interpolation of the rain gauges is widely used, although the use of radar and satellite data are increasingly widespread. Different data sources are needed to correct the spatial errors (that can affect the rain gauges measurements) due to variability in precipitation over short distances. Since these errors are related to the density and distribution of rain gauges, they affect the performance of the model streamflow simulation. As a matter
of fact, the spatial distribution of the rain gauge monitoring network may not respect a minimum density standard established by the WMO, as in the case of the selected domain, particularly over the complex orography areas. Therefore, in the areas where rain gauges are very limited or even absent, the inclusion of modern remote-sensing techniques such as radar or satellite data is essential.

The study highlights the benefit of using Satellite-based RainFall Estimation precipitation products for
hydrological simulations, especially in those areas where there is no homogeneous distribution of rain gauges. With these



purposes, the Cellular Automata based algorithm has been developed and implemented in the CHyM code, an application of advanced assimilation, interpolation, and merging techniques. This work supports the relevance and provides methodologies, for dealing with different data sources simultaneously (rain gauges and satellite rainfall estimates): different data sources are used to obtain a mutual correction of the implicit error typical of different sources. An important

aspect is choosing the right methodology for using the data. The main aim of this work is to validate the CA technique as a tool for creating APE using rain gauge and satellite rainfall data. The temporal resolution of these data sources (rain gauge is provided every hour instead of satellite data every half hour) is essential for developing operational monitoring and forecasting tools for flood early warning systems.

Eight different simulations have been carried out, where the hydrological model has been forced with different

CA-based APE scenarios. UNCAL and CAL simulations use only satellite data respectively IMERG-F Uncalibrated and IMERG-F Calibrated, GAUGE simulation uses rain gauge data; the GAUGEUNCAL simulation uses the combined gauge and satellite data using the NoModular approach, a high-resolution lattice is filled with both the satellite and the rain gauge data, used simultaneously at each time step, prioritizing the rain gauge data. MODGAUGEUNCAL1, MODGAUGEUNCAL3, MODGAUGEUNCAL5 are the simulations where the hydrological model has been forced

using a Modular approach, a hierarchical sequence of modules is used to assimilate the different data sets to fill the high-resolution lattice. Different radii of influence related to rain gauge data (the number at the end of the simulation name is related to the gauge radius of influence: 1km, 3km and 5km), merging gauge and Uncalibrated satellite data, whereas the last hydrological simulation, MODGAUGECAL5 is carried out in the same way, but using the Calibrated satellite data.

For the comparison between observed and simulated stream flow time series, objective functions were selected.

The traditional performance indicators have been used and, in addition, the typical signal theory indicators were selected to obtain a more objective analysis, independent of the limits of the scores commonly used for hydrological analyses. The results show an improvement in the performance of hydrological simulations when satellite and rain gauge data are merged. All scores confirm the better performance using the only rain gauge data (GAUGE) compared to satellite data (UNCAL, CAL), but the best performances are associated to the model outputs forced with APE obtained starting from

a rain gauge background rain field characterized by radius of influence $R = 5km$ (i.e., when 68% coverage of the Tanaro basin is associated to the rainfall estimated through the gauge data) and the remaining part of the area is covered by the rainfall field rebuilt using the GPM IMERG product (calibrated or uncalibrated). This study is the first detailed analysis of the potential usage of the CA technique to merge different rainfall data inputs to hydrological models. In the future, this method will be tested on a larger number of case studies and different river basins, as well as on other satellite products

(available at different spatial, temporal resolution and shorter latency) to investigate the advantage of the proposed approach in an operational setting for near-real time hydrological applications.



**Data availability**

All raw data can be provided by the corresponding authors upon request.

DEM data are accessibile at https://land.copernicus.eu/imagery-in-situ/eu-dem/ (last access August 2023).

The fifth generation ECMWF reanalysis (ERA5) are accessibile at https://cds.climate.copernicus.eu/cdsapp#!/dataset/reanalysis-era5-pressure-levels?tab=form (last access July 2023).

Drainage Network and boundaries basin are accessible at https://www.hydrosheds.org/ (last access August 2023).

**Author contributions**

AL, BT, VC and GP planned the work; AL, BT, LDA processed the data; AL, BT and VC analysed the results; AL and BT wrote the manuscript draft; GP, VC, FSM, GR, LDA, RL, MM, and PT reviewed and edited the manuscript.

**Competing interests**

The authors declare that they have no conflict of interest.

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





**Figure 1**: North-West domain of Italy, Tanaro river basin is hilighted in yellow. The numbers represent the basin flow discharge stations selected: Montecastello (7956 km$^2$ drained), Masio (4535 km$^2$ drained), Asti (4123 km$^2$ drained), Alba (3385 km$^2$ drained), Piantorre (500 km$^2$ drained), Mondovì – Ellero (180 km$^2$ drained), San Damiano d'Asti – Borbore (85 km$^2$ drained), Ponte di Nava (149 km$^2$ drained). The red triangles are the rain gauges available for this study. DEM data are accessibile at https://land.copernicus.eu/imagery-in-situ/eu-dem/ (last access August 2023). Drainage Network and boundaries basin are accessible at https://www.hydrosheds.org/ (last access August 2023).






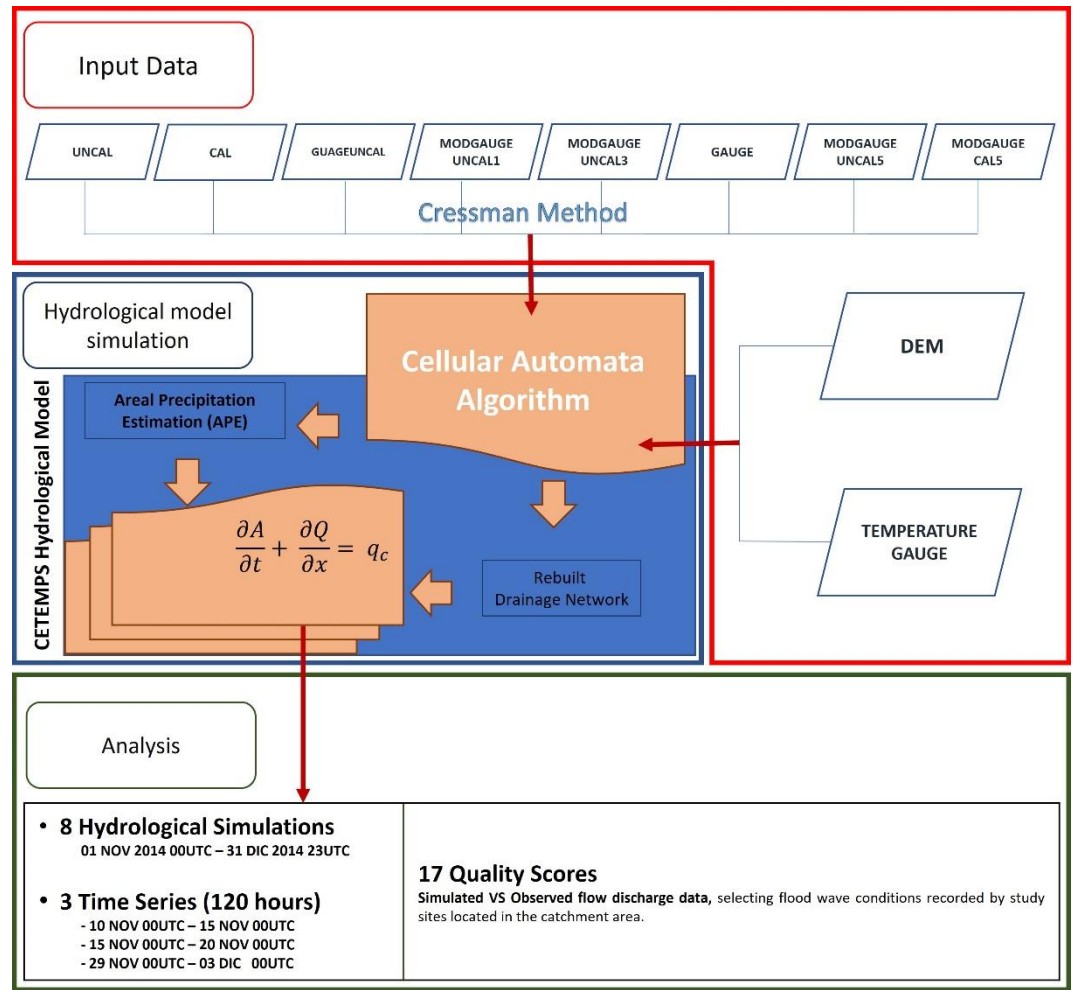

**Figure 2**. Numerical experiment workflow, three main tasks are selected: 1) precipitation data gridding, 2) precipitation data assimilation and merging, 3) hydrological model simulation and error score calculation. Different combinations of precipitation are tested as input to the hydrological model and error scores calculated accordingly in terms of flow discharge. Eight different simulations have been carried out for each case studies, using the eight different rain input setting.







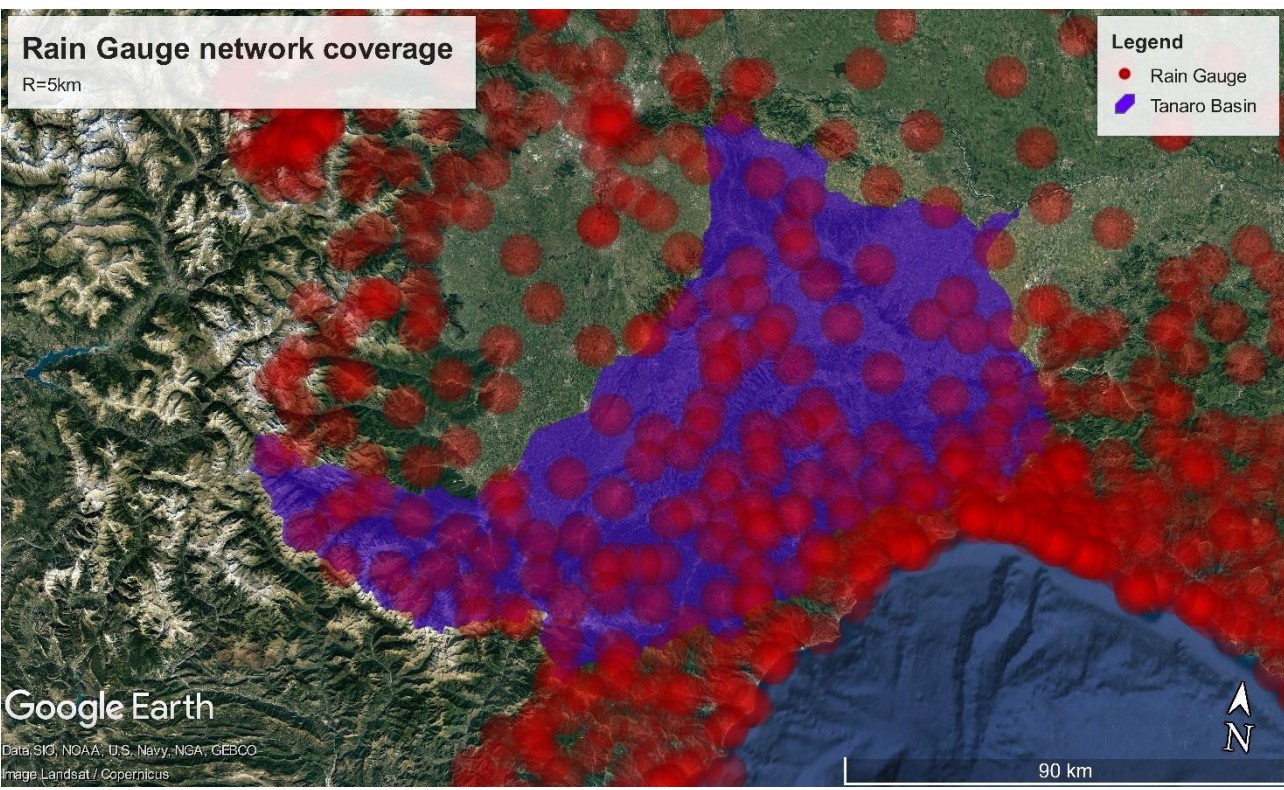

**Figure 3**. Tanaro Basin rain gauge density and distribution. The red circles represents the rain gauge coverage of area using a radius of influence of 5 km. The blue area reprents the Tanaro basin extent. © Google Earth.







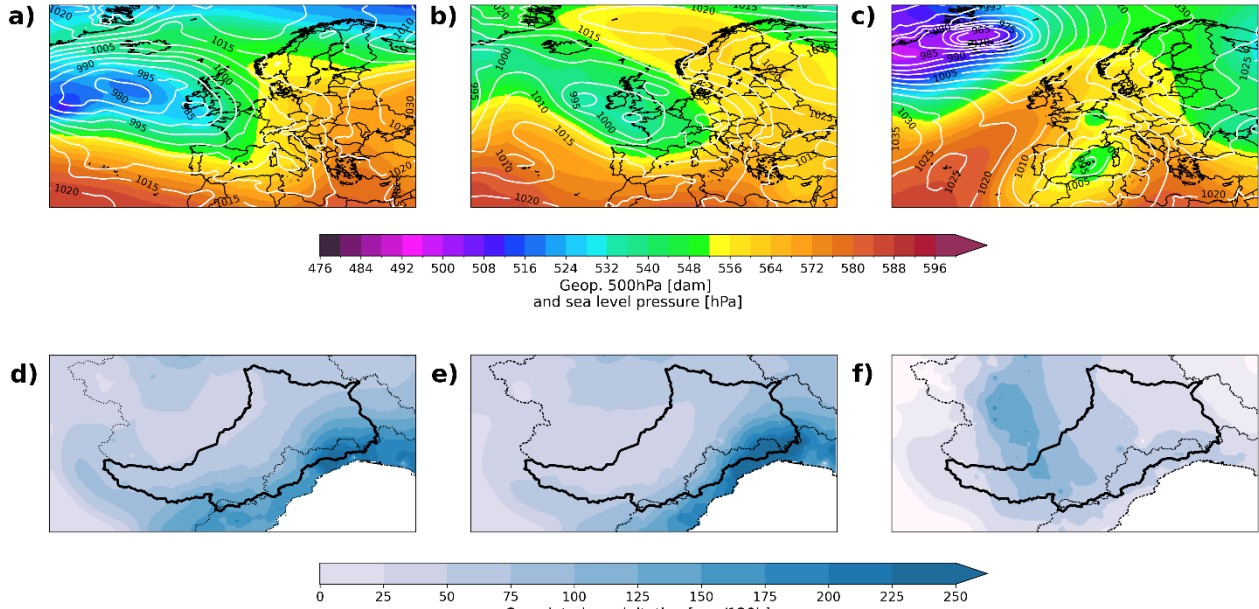

**Figure 4**. Case Studies Synoptic analysis: a) CS 01 12 November 2014 00UTC, b) CS 02 16 November 2014 00UTC, c) CS 03(1 December 2014 00UTC) 500 hPa geopotential height and sea level pressure using the fifth generation ECMWF reanalysis (ERA5); d) CS 01, e) CS 02, f) CS 03 120h cumulated rain rebuilt using rain gauge data.


**Figure 5**: CS 01 Areal Precipitation Estimation: the figures show the rain field rebuilt using rain gauge data (a), GPM IMERGE FINAL CAL (b), GPM IMERGE FINAL UNCAL (c). Panel d shows the rain field obtained forcing the hydrological model with rain gauge data, using a radius of influence equal to 5km, merged with GPM IMERGE FINAL UNCAL.


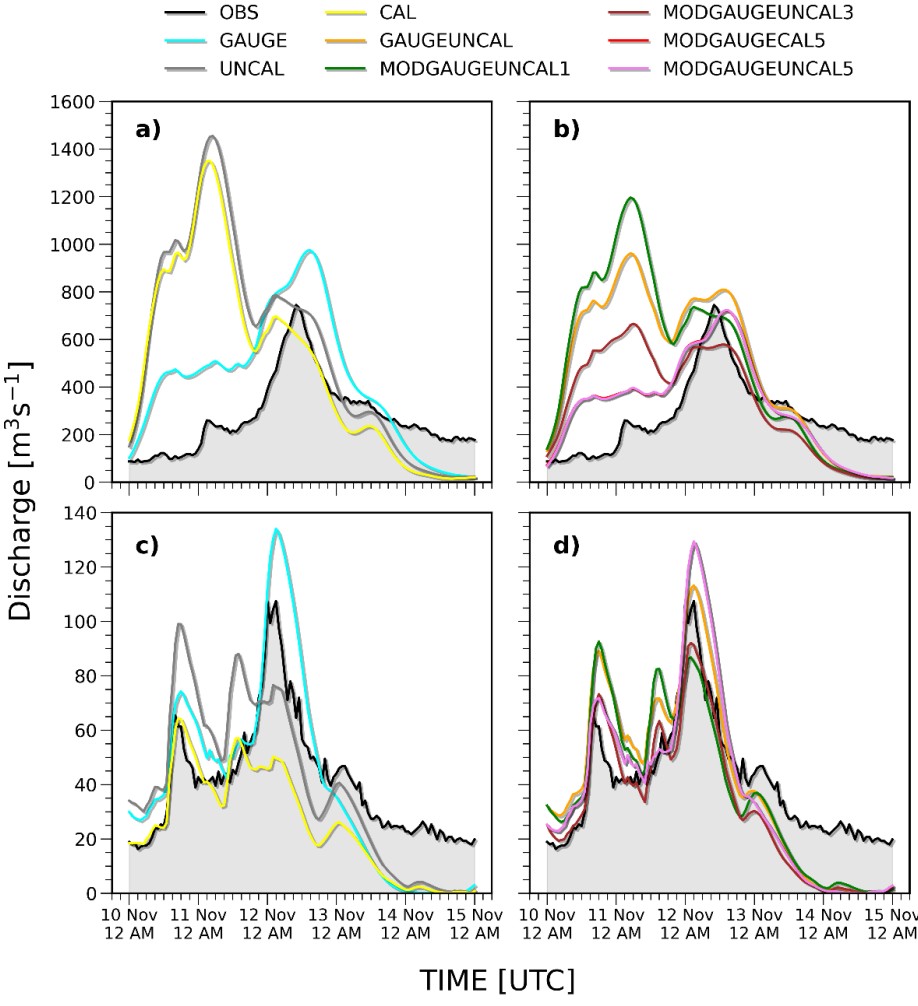

**Figure. 6**. Intercomparison between Observed and Simulated flow discharge data with the different rainfall scenarios for CS01. The Simulation analysis are related to Alba Tanaro (a, b) and Ponte di Nava (c, d) river sections.



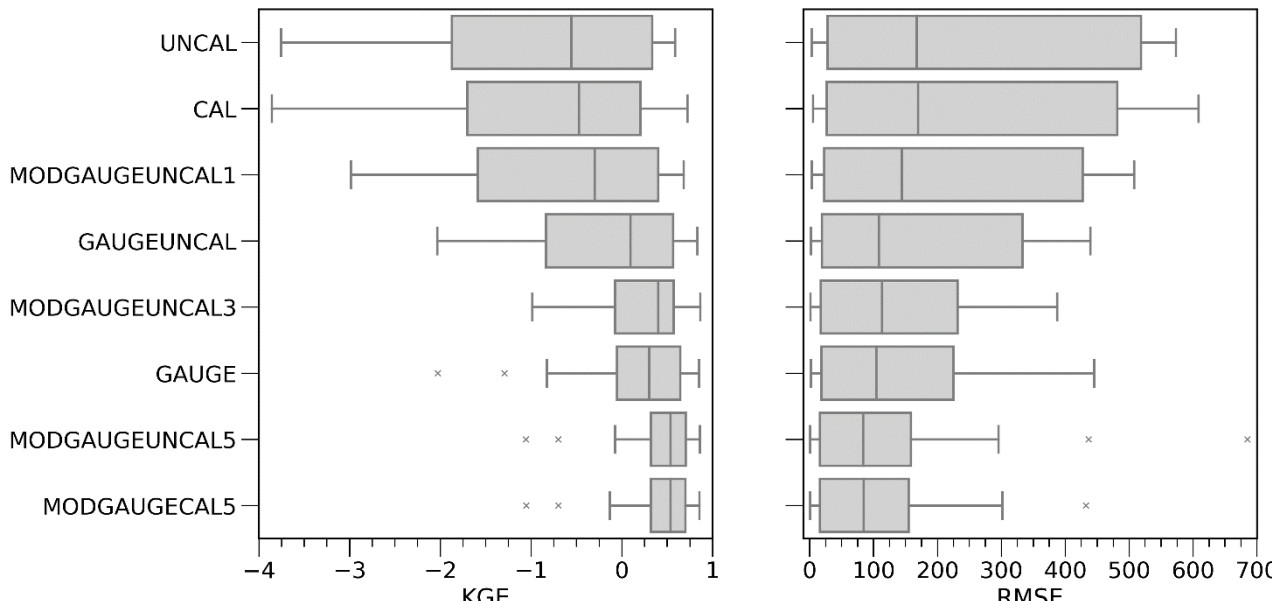

**Figure 7**. The boxplots show the summary refer to KGE and RMSE obtained from the CHYM simulation using the eight APE fields as input, related to all three Case Studies and all river sections. The rain gauge data (GAUGE simulation) allows for better performance than using the satellite data alone (UNCAL and CAL), but the results are better if the two data sources are merged, especially using the MODGAUGECAL5 setting, which is comparable to MODGAUGEUNCAL5. The same results are obtained both in terms of quality score (KGE) and error estimation (RSME)

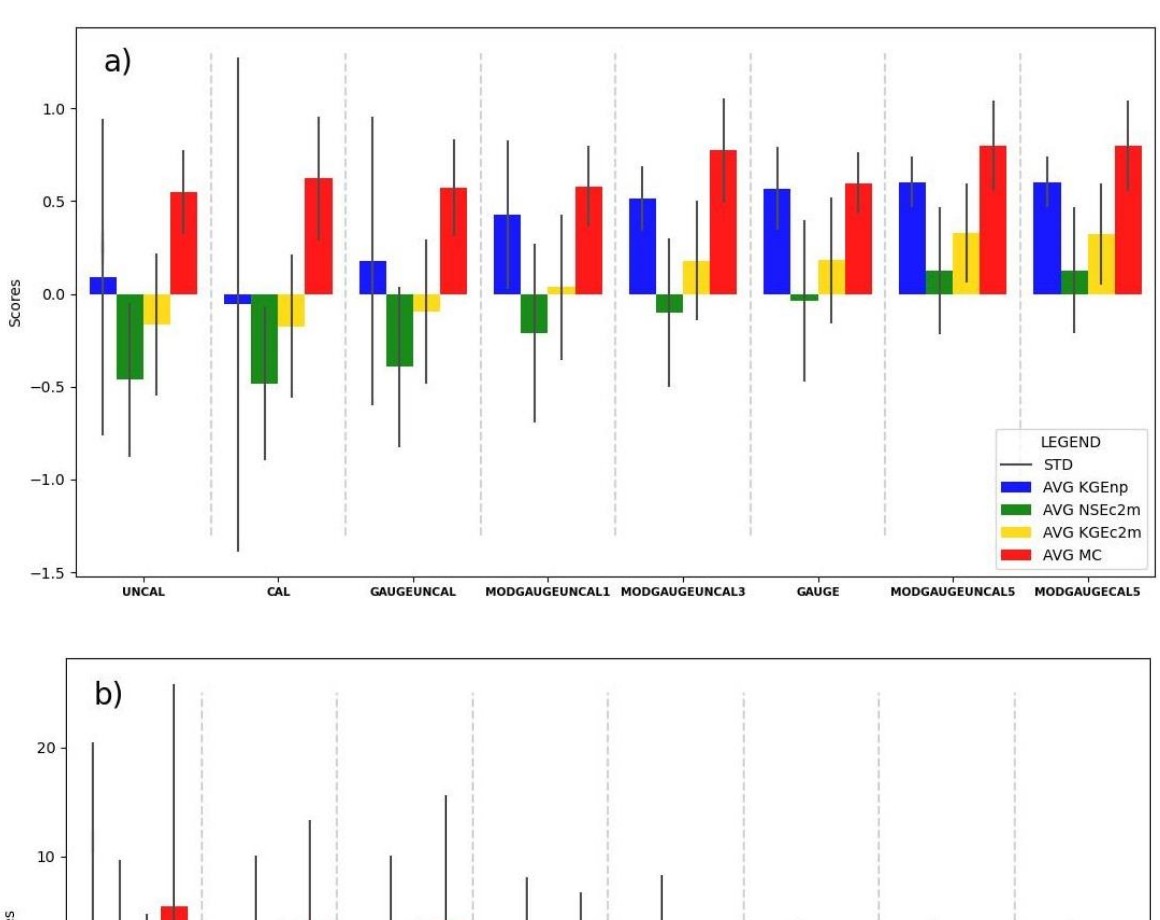

**Figure 8**. The histograms summarize the statistical analysis performed for the different scores, evaluating their average (AVG) and the standard deviation (STD). The figure a) shows the quality scores (KGEnp, NSEc2m, KGEc2m and MC) where the best performances are identified by a value equal to 1. In the figure b) (MARE, CT_D, E% and DDTW) the best performances are identified by values close to zero.




**Table 1**. Raingauge Network characteristic of Tanaro catchment; $R^*$ = Cressman radius of influence.

| Region Type | Area size (km²) | Network characteristics | | Average Gauge Distance (km) | Gauge Covered Area (R=5km) * | Gauge Covered Area (R=3km) * | Gauge Covered Area (R=1km) * |
|---|---|---|---|---|---|---|---|
| | | Gauge Numbers | Network Density (km² per gauge) | | | | |
| Mountain (>700 MSL) | 2241 | 22 | 102 | 10 | 0.79 | 0.26 | 0.006 |
| Hill (700 ≤ H ≤ 300) | 3032 | 32 | 95 | 9.74 | 0.74 | 0.26 | 0.008 |
| Flat (< 300) | 3153 | 19 | 166 | 13 | 0.51 | 0.16 | 0.004 |
| **Tanaro Catchment** | **8426** | **73** | **115** | **10.74** | **0.68** | **0.22** | **0.006** |






**Table 2**. Average statistical scores for all three case studies and all river stations obtained from the CHYM simulations using the eight APE fields as input (AVG). The first block of the table shows all the quality scores where the best performances are identified by a value equal to 1. In the second block of the table, the best performances are identified by values close to zero.


| | UNCAL | CAL | MOD GAUGE UNCAL1 | GAUGE UNCAL | MOD GAUGE UNCAL3 | GAUGE | MOD GAUGE UNCAL5 | MOD GAUGE CAL5 |
|---|---|---|---|---|---|---|---|---|
| KGE | -1.414 | -1.425 | -1.029 | -0.408 | 0.069 | 0.116 | 0.407 | 0.403 |
| NSE | -15.618 | -12.739 | -10.425 | -4.570 | -1.775 | -0.997 | -0.104 | -0.104 |
| KGEprime | -0.711 | -0.989 | -0.565 | -0.139 | -0.070 | 0.186 | 0.242 | 0.239 |
| KGEnp | 0.090 | -0.056 | 0.178 | 0.427 | 0.514 | 0.568 | 0.604 | 0.604 |
| NSEc2m | -0.462 | -0.482 | -0.392 | -0.210 | -0.101 | -0.037 | 0.126 | 0.128 |
| KGEc2m | -0.164 | -0.175 | -0.097 | 0.036 | 0.179 | 0.181 | 0.327 | 0.325 |
| KGEprime_c2m | -0.150 | -0.199 | -0.112 | 0.034 | 0.069 | 0.202 | 0.251 | 0.250 |
| KGEnp_c2m | 0.173 | 0.156 | 0.218 | 0.336 | 0.363 | 0.428 | 0.446 | 0.447 |
| MC | 0.548 | 0.622 | 0.572 | 0.579 | 0.774 | 0.598 | 0.798 | 0.797 |
| | | | | | | | | |
| RMSE | 297.038 | 277.595 | 257.797 | 200.080 | 159.440 | 149.397 | 128.757 | 129.549 |
| MARE | 1.163 | 1.299 | 1.055 | 0.778 | 0.630 | 0.586 | 0.464 | 0.464 |
| PBIAS | -53.840 | -57.306 | -43.956 | -25.297 | 8.220 | -13.323 | 15.662 | 15.383 |
| CT_D | -2.750 | -2.750 | -2.125 | -1.042 | -1.042 | 0.750 | 0.625 | 0.625 |
| TP_D | -4.083 | -4.833 | -3.375 | -1.958 | -1.875 | 3.875 | 3.750 | 3.792 |
| E% | 1.543 | 1.318 | 1.176 | 0.716 | 0.261 | 0.275 | -0.021 | -0.019 |
| DTW | 50.829 | 40.878 | 33.380 | 15.190 | 6.978 | 4.346 | 2.674 | 2.660 |
| DDTW | 5.436 | 3.043 | 3.274 | 1.481 | 0.668 | 0.257 | 0.192 | 0.192 |






**Table 3**. Standard deviation of the statistical scores for all three case studies and all river stations obtained from the CHYM simulations using the eight APE fields as input (STD).


| | UNCAL | CAL | MOD GAUGE UNCAL1 | GAUGE UNCAL | MOD GAUGE UNCAL3 | GAUGE | MOD GAUGE UNCAL5 | MOD GAUGE CAL5 |
|---|---|---|---|---|---|---|---|---|
| **KGE** | 2.914 | 2.617 | 2.386 | 1.536 | 1.003 | 0.701 | 0.445 | 0.448 |
| **NSE** | 42.912 | 25.613 | 28.062 | 12.344 | 5.752 | 2.480 | 1.226 | 1.228 |
| **KGEprime** | 1.126 | 1.439 | 1.029 | 0.795 | 0.802 | 0.609 | 0.634 | 0.637 |
| **KGEnp** | 0.854 | 1.333 | 0.779 | 0.402 | 0.173 | 0.224 | 0.138 | 0.137 |
| **NSEc2m** | 0.414 | 0.415 | 0.433 | 0.482 | 0.401 | 0.435 | 0.340 | 0.341 |
| **KGEc2m** | 0.383 | 0.387 | 0.388 | 0.392 | 0.321 | 0.342 | 0.268 | 0.272 |
| **KGEprime_c2m** | 0.298 | 0.315 | 0.298 | 0.298 | 0.297 | 0.316 | 0.332 | 0.333 |
| **KGEnp_c2m** | 0.313 | 0.327 | 0.306 | 0.261 | 0.153 | 0.200 | 0.143 | 0.142 |
| **MC** | 0.266 | 0.335 | 0.260 | 0.217 | 0.282 | 0.166 | 0.242 | 0.243 |
| | | | | | | | | |
| **RMSE** | 366.182 | 289.936 | 324.530 | 244.503 | 191.808 | 167.846 | 159.119 | 164.644 |
| **MARE** | 0.984 | 1.437 | 0.898 | 0.521 | 0.318 | 0.306 | 0.186 | 0.187 |
| **PBIAS** | 101.932 | 151.940 | 94.248 | 53.677 | 34.796 | 34.535 | 23.365 | 23.417 |
| **CT_D** | 12.477 | 12.804 | 12.221 | 9.154 | 9.370 | 2.933 | 3.080 | 3.107 |
| **TP_D** | 14.180 | 14.020 | 14.041 | 12.614 | 12.204 | 3.756 | 3.711 | 3.730 |
| **E%** | 3.166 | 2.295 | 2.451 | 1.650 | 1.101 | 0.640 | 0.533 | 0.533 |
| **DTW** | 126.770 | 70.688 | 81.781 | 34.897 | 15.763 | 6.349 | 3.024 | 3.020 |
| **DDTW** | 20.408 | 10.338 | 12.362 | 5.263 | 2.326 | 0.562 | 0.371 | 0.372 |



**Table 4**. Median of the statistical scores for all three case studies and all river stations obtained from the CHYM simulations using the eight APE fields as input (MED).

| | UNCAL | CAL | MOD GAUGE UNCAL1 | GAUGE UNCAL | MOD GAUGE UNCAL3 | GAUGE | MOD GAUGE UNCAL5 | MOD GAUGE CAL5 |
|---|---|---|---|---|---|---|---|---|
| **KGE** | -0.558 | -0.473 | -0.299 | 0.097 | 0.402 | 0.299 | 0.537 | 0.537 |
| **NSE** | -3.140 | -3.609 | -2.083 | -0.637 | -0.264 | 0.046 | 0.341 | 0.331 |
| **KGEprime** | -0.485 | -0.585 | -0.308 | 0.030 | 0.116 | 0.362 | 0.532 | 0.529 |
| **KGEnp** | 0.392 | 0.375 | 0.500 | 0.606 | 0.523 | 0.657 | 0.611 | 0.619 |
| **NSEc2m** | -0.604 | -0.643 | -0.508 | -0.240 | -0.116 | 0.028 | 0.207 | 0.199 |
| **KGEc2m** | -0.218 | -0.182 | -0.129 | 0.051 | 0.251 | 0.176 | 0.368 | 0.367 |
| **KGEprime_c2m** | -0.195 | -0.225 | -0.133 | 0.015 | 0.062 | 0.221 | 0.362 | 0.360 |
| **KGEnp_c2m** | 0.244 | 0.231 | 0.333 | 0.435 | 0.354 | 0.489 | 0.440 | 0.448 |
| | | | | | | | | |
| **RMSE** | 167.351 | 169.794 | 144.205 | 108.312 | 113.309 | 104.219 | 83.801 | 84.676 |
| **MARE** | 0.915 | 0.845 | 0.817 | 0.635 | 0.531 | 0.533 | 0.457 | 0.467 |
| **PBIAS** | -30.560 | -20.430 | -17.750 | -4.952 | 20.578 | -1.279 | 20.597 | 21.030 |
| **CT_D** | 0.535 | 0.560 | 0.570 | 0.585 | 0.765 | 0.570 | 0.750 | 0.750 |
| **TP_D** | 2.000 | 1.000 | 1.500 | 1.000 | 1.000 | 0.000 | 0.500 | 0.500 |
| **E%** | 0.100 | 0.500 | 1.100 | 2.200 | 2.100 | 4.000 | 3.500 | 3.500 |
| **DTW** | 0.630 | 0.605 | 0.605 | 0.230 | -0.075 | 0.075 | -0.220 | -0.220 |
| **DDTW** | 9.940 | 11.605 | 7.000 | 2.255 | 3.020 | 1.945 | 2.000 | 2.005 |
| **RMSE** | 0.140 | 0.145 | 0.125 | 0.065 | 0.045 | 0.040 | 0.040 | 0.040 |
