# Peer review of "On the combined use of rain gauges and GPM IMERG satellite rainfall products for the hydrological modelling: impact assessment of the cellular automata-based methodology on the Tanaro river basin in Italy."

_Hydrology and Earth System Sciences, 2023_

## Author Comment (AC1)

First, we would like to thank the reviewers for having carefully read the paper and provided valuable comments which helped us to improve the quality of the manuscript. We have taken into consideration all the comments raised by the reviewers and changed the manuscript accordingly. The details of our changes are highlighted in the main text. The point-by-point answers to Reviewers #1 and #2 are provided in the following and highlighted in red.

**RC1**:'Comment on hess-2023-214', Anonymous Referee #1, 09 Oct 2023

**General Comments**

The authors tackle an important topic, namely how to best combine gauge and satellite precipitation estimates for applications. The hydrological validation they pursue is a reasonable way of testing how the various input datasets perform, and this is clearly the strong point of the manuscript. As such, the existing examples and conclusions related to hydrology seem solid.

The other big point of the manuscript, as the title makes clear, is the combination scheme that employs CA interpolation, but here the manuscript falls short. I would expect a step-by-step demonstration that all the extra mathematical complication produces precipitation fields that are physically meaningful and more consistent with the input fields than some simpler scheme. I would consider it mandatory to address this issue.

The manuscript has been revised. A new figure 4 is added to show the physical consistency of the combination scheme that employs CA.

The English is not ready for publication; I have commented on a few word choices that I found confusing or that might escape a general technical editor, but not otherwise. Other issues needing attention are listed below.

If accepted for publication, we will make use of the English editing service of the journal.

**Specific Comments**

1. ***CA and regularly gridded data:*** *It is not clear to me how CA handles the regularly gridded data. Does it assign each satellite gridbox value to the finer gridbox closest to that satellite gridbox's center? Conventionally, gridded data give you an average value across the entire box, so if you just assign that value to the box's center, doing an interpolation and then averaging that field back to the original resolution will not give you the right average, in general.*

We Thank the reviewer for her/his comment. The technique implemented does not pose any constraint on the final interpolated field (i.e., averaging the output field back to the original resolution will not likely give the initial average value exactly). Indeed, what we did is to redistribute spatially the rain information from multiple sources, namely, rain gauges and satellite retrievals. For the former, the measured precipitation is assigned to the closest grid point in our destination grid and then interpolated over the neighbourhood grid points using Cressman inverse distance interpolator with a ray of influence, for example, of 5 km. From the satellite standpoint, the rainfall values in each Field Of View (FOV) of the satellite are first assigned to the respective FOV center. Secondly, each FOV center is assigned to the closest grind point in the destination grid as done for the rain gauges. Thirdly, each satellite rainfall value thus obtained is interpolated with Cressman inverse distance with a ray of influence equal to the native FOV size. So doing, the areal information from the satellite is spatially redistributed with the inverse distance criteria. Although this can sound a little bit arbitrary because we do not really know how precipitation is distributed within each satellite FOV, the methodology that we implemented has the advantage to treat both rain gauges and satellites equally in terms of the processing applied. In addition, the approach proposed follows a source hierarchy, for MODULAR approach, which means that rain gauges are interpolated first and then the satellite comes into play for gap filling only, and in the NOMODULAR approach, both satellite and rain gauge data are interpolated simultaneously at each time step. Eventually, the Cellular Automa has the role to make the final reconstructed rainfall field spatially consistent. Note that the goal of the manuscript is not to propose a new multi-source merging strategy, but instead to demonstrate how the use of multiple sources can improve the hydrological output.

For a more in-depth description of the technique, in Coppola et al 2007 you find different examples of applying Modular approach sequences with two and three modules where the rain sources utilized in each module are detailed.

*Specifically, locally convex-up areas will be underestimated (particularly sharp peaks), concave-up areas will be overestimated, and the boundary of the precipitating region will spread somewhat into the non-raining area. I consider discussion of these issues to be mandatory.*

Certainly, locally convex-up areas will be underestimated (particularly sharp peaks), concave-up areas will be overestimated, and the boundary of the precipitating region will spread somewhat into the non-raining area. However, our results indicates that the distribution of input precipitation field with smoothed peaks and minima but with a more homogeneous spread, improves the estimate of the flow discharge and peak timing in the hydrological model output.

2. *What is the model grid spacing:* Section 4.3 seems closest to stating this, but I missed seeing a declarative statement giving the specific value of the grid spacing. If you're really down at "hundreds of meters", it needs to be made clear that the effective resolution of the precipitation data is back at 5, 10, or more km. Finer than that is just

more and more precisely defining smooth variations between the available value locations.

A detailed comment has been inserted on lines L290-293. The hydrological spatial resolution is approximately 900 m, the same as used for the rainfall field. The approach was explained in the previous comment.

3. *Example interpolated precipitation fields:* Given the emphasis on the innovative combination of data sources, I would expect to see a sequence of maps illustrating the process and improvement versus simpler "traditional" schemes. This should happen first, before pointing to the accumulated precipitation in Fig. 5 and the aggregate hydrological results in Fig. 6 (for example). This aspect of the manuscript is where the issues in item 1 need to be addressed.

Thanks for this tip, this is a great idea for future work. Note that the priority of this study is to demonstrate how the use of combined data improves the hydrological model's output. Probably, this can be achieved even using "traditional" data merging schemes. This is the reason why comparisons with other assimilation techniques are not implemented. Figures 6 and 7 are useful to demonstrate that the different data sources give different rainfall fields to the point of improving or worsening the results of the hydrological simulation.

4. *Boundary bias:* This phrase comes up several times, but it was never quite defined as to what boundary was being discussed. It is stated that data outside the basin is used, which presumably should solve a problem at the basin boundary. Perhaps the problem is along the southwest side of the basin, where no additional gauges are shown in Fig. 1. The statement in L.417-418 should appear a lot sooner in the text and be more explicit about how this works.

In general, boundary bias denotes a systematic error or distortion occurring in data or models near the periphery of a study area or dataset. This bias often arises due to differing conditions or factors at the edges of the study area compared to its interior. The presence of boundary bias can impact the accuracy and reliability of analyses and interpretations, underscoring the importance of acknowledging and addressing this bias when working with spatial data or models to ensure more precise and meaningful outcomes.

In the specific context of this study, boundary bias manifests as a discontinuity associated with the absence of observed data beyond the confines of the basin. To mitigate this potential issue, whenever feasible, all available data across the entire domain is utilized. Nevertheless, a discontinuity persists since no observed data beyond the Italian border are accessible. This aspect will be further discussed later in the paper.

To further elucidate this concept, the following sentence was incorporated into lines L229-234: "Furthermore, a strategy used by the work to avoid boundary effects is to

extend the spatial domain well beyond the studied basin: this strategy is useful for a better reconstruction of the precipitation field (Figure 1). Many data used, although redundant, lead to a better reconstruction of the rain field. A smaller amount of this data would probably be enough, but the work uses everything that the national rain gauge network has available. Future studies could lead to identifying, given their distribution, enough rain gauges outside the basin deemed useful to overcome the boundary effect."

5. *Fig. 1: I question whether all those gauges outside the basin are really useful and therefore worth depicting. I would suggest that as you move outside the boundary you can stop after you pass about 3 gauges (which of course varies with coverage). Was it really not possible to obtain gauge data to the southwest of the basin? This introduces the boundary effects the manuscript discusses (right?).*

   The aim of this work is to demonstrate how to validate an operational chain for civil protection monitoring and forecast purposes. The rain gauge network that was used is the official national one. Using all the data at our disposal, beyond a defined number (3 or 4) outside the basin for example, helps the model to better distribute the rainfall field, it is certainly not a limit, but rather an added value. A possible future work could be to estimate the optimal number of rain gauges, based on their distribution.

   So, the idea is to use all the rain gauge data available for a possible operational activity, which does not include those outside the national border. These limitations help our validation: despite the many critical issues the model seems to respond very well.

6. *Fig. 7 caption:* The statements after the first sentence are interpretation and belong in the text.
   The sentence has been deleted.

**Technical Corrections**

1. *L.180, 720:* "O" is actually that author's last name; "Sungmin" is her first name. Done. Thanks for your suggestions.

2. *Nine occurrences:* "IMERG" is mis-stated as "IMERGE". Done. Thanks for your suggestions.

3. *L.52:* Not sure "captative" is the right word. Done. Thanks for your suggestions.

4. *L.88-89:* Awkward phrasing, including that "peculiarity" is probably something like "availability". Done. Thanks for your suggestions.

5. *L.168 and 3 other locations:* I'm not sure what "rain bandwidth" means.

Shi et al. (2020): "any region has an effective influence (ERI) radius for any rainfall event, which reflects the influence of a rain bandwidth, and such an EIR is not larger than a certain distance". The reference to L269 is wrong.

6.  *L.186:* Should this be "…that are usually not instrumented."? Done. Thanks for your suggestions.

7.  *L.193:* Fig. 2 refers to a "workflow", not "rationale", which seems better. Done. Thanks for your suggestions.

8.  *L.193:* Fig. 2 says three tasks, not four. Done. Thanks for your suggestions.

9.  *L.202*: Unclear phrasing; are you saying something like "the model is not calibrated specifically for this study's cases"? The reviewer is right; this sentence is misleading. It was meant that the model was calibrated in past studies on the Po basin of which the Tanaro River is a tributary (Coppola et al 2014) and not for this study. To avoid confusion, it has been deleted from the text.

10. *L.428:* I think "intuitive" could better be "subjective". Done. Now Line 446

11. *Fig. 1:* a) What are the blue lines? b) The hydrometers are nearly invisible; maybe they should be plotted in white? Done. The description of the blue lines has been added in the caption. The figure has been replaced.

12. *Fig. 2 caption:* The phrasing is awkward, perhaps something like "… workflow, consisting of three main tasks:" Done. Thanks for your suggestions.

13. *Fig. 3*: The mostly-dark colors make it really hard to distinguish the basin and gauge coverages. I'd say the Google Earth background and the basin's blue need to be much lighter. Done. A new figure is available.

---

## Author Comment (AC2)

First, we would like to thank the reviewers for having carefully read the paper and provided valuable comments which helped us to improve the quality of the manuscript. We have taken into consideration all the comments raised by the reviewers and changed the manuscript accordingly. The details of our changes are highlighted in the main text. The point-by-point answers to Reviewers #1 and #2 are provided in the following and highlighted in red.

**RC2**: 'Comment on hess-2023-214', Anonymous Referee #2, 11 Oct 2023

The paper of Lombardi et al. deals with an interesting topic concerning the reliability and usability of satellite rainfall products for hydrological applications. To this aim, they consider a case study in northern Italy (Tanaro watershed) and test different precipitation fields achieved using only rain gauges, satellites, or several merging options. Precipitation fields are used as inputs for a hydrological model. According to the behaviour of the resulting hydrograph, the precipitation fields are judged in terms of reliability, supporting the discussion with several (very many, indeed) skill scores.

Despite the interest in the topic, I believe several changes in the paper are needed before it can be considered ready for publication in HESS.

First, I have concerns about the methodology followed. As a first step, since the main point is the reliability of the satellite rainfall products in providing quantitative precipitation estimates, I would have set up a validation exercise with observed (rain gauge) precipitation using, e.g., a leave-one-out or cross-validation approach.

> Thanks for the comment, it was intentionally decided not to use this approach, relying exclusively on an indirect method of validation through a hydrological evaluation. The work does not aim to do a local validation but a distributed one, thanks to the analysis of data relating to the drainage network.

Then, an indirect validation through simulated hydrographs should be justified more strongly, mainly if performed with a not-calibrated hydrological model (L202). This approach could be tricky and misleading because of the inherent limitations of the non-calibrated hydrological model so that errors can counterbalance and smooth each other. I suggest a preliminary calibration of the hydrological model with a reference precipitation field (e.g., only rain gauges) and only afterwards assess the changes caused to the modelled hydrograph with other methods.

> The reviewer is right; this sentence is misleading. It was meant that the model was calibrated in past studies on the Po basin of which the Tanaro River is a tributary (Coppola et al 2014) and not for this study. To avoid confusion, it has been deleted from the text.

Another concern is the description of the CA-based technique for interpolating and merging precipitation, which is unclear to me. An example should be given. Among other things, it is unclear why the authors also consider the time evolution of the precipitation field.

Regarding the technique adopted in detail, the review can refer to Coppola et al, 2010. Effectively, a detailed description of the CA algorithm is missing in the manuscript and the following section has been added.

The temporal evolution of the rainfall field is fundamental for the hydrological simulation, in particular, the CHyM model is forced with the hourly rainfall fields.

Last but not least, the paper could be structured much better.

The reviewer is right; the document has been rearranged.

The different options for achieving the precipitation fields are not clearly presented (e.g., some acronyms are provided in Fig. 2, then they are explained much later -L400- and not in the Methods, but in the Results section).

A description table has been added (table 2)

It is unclear why the authors decided to rely on 17 scores. This choice is somewhat confusing, in my opinion.

The 17 scores are necessary to obtain an objective evaluation of the analysis, given that each score highlights different characteristics. Obviously, if they are all positive, the results are confirmed as reliable.

Furthermore, many of the methods are presented in the results section or even in the conclusions (please refer to specific comments below). The discussion should refer to similar analyses performed by other authors to contextualise the results better. The conclusions section should be more than a summary of the paper.

In summary, I saw the possible added value that this paper can bring to the scientific community. Still, a thorough review is needed regarding the methodological approach and the structuring of the article. Please find below some other minor to moderate comments. I hope my review helps improve the quality of the paper.

1. *LL46-49: please revise. Dealing with predicted rainfall, it's impossible to remove uncertainty (correctly, in fact, the second sentence of the paragraph refers only to observed rainfall data). I suggest focusing on why accurate spatial distribution of rainfall observation is important.*

To avoid confusion, the sentence has been corrected as follows: "As far as the operational activity is concerned, the hydrological models are usually forced both with observed and forecasted rainfall data, and the uncertainty of hydrological forecasts is strongly related to the uncertainty of the input rain field. Therefore, providing hydrological models with observed precipitation data that is as realistic as possible becomes essential

in mitigating uncertainty, during the spin-up phase of the simulation when the hydrological model is forced with observed rainfall data."

> 2. *L104: "The work": do the authors refer to Shi et al. (2020)?*

Yes. The sentence has been corrected.

> 3. *LL111-112: I can't entirely agree with this statement. Indeed, there are a lot of studies dealing with this topic.*

Done. The sentence has been deleted.

> 4. *LL118-119 is a repetition of the second main objective declared at LL115-116. If the authors agree with my comment, please consider if the previous sentence (LL117-118) is well-placed and contextualised.*

The reviewer is correct; the second sentence LL117-118 has been deleted.

> 5. *L120: are all these 352 stations really useful? I guess the authors only need those lying into or close to the analysed watershed. From this point of view, it's unclear why the authors consider a much broader spatial domain than the investigated watershed (which, moreover, is not at the centre of the domain itself). I guess many stations, for example, lying in the north and northwest, are useless for this case study.*

The objective of this study is to validate a prospective operational framework designed for civil protection monitoring and forecasting. The rain gauge network employed in this research is the official national network. Utilizing all available data enhances the model's ability to effectively distribute the rainfall field, representing not a limitation but rather an added value. As rightly noted by the reviewer, the inclusion of 352 rain gauge data may be excessive, especially considering that only 73 of these data points fall within the specific basin under consideration. At the state of the art, a study aimed at understanding how much rain gauge data is sufficient to have the same results has not yet been carried out A possible future work could be to estimate the optimal number of rain gauges, based on their distribution.

It is crucial to acknowledge that the algorithm employed in this study applies smoothing to the processed data. Therefore, for scientific rigor, the entire operational domain is utilized, acknowledging both its advantages and limitations. One such limitation is the absence of rain gauge data beyond national borders, which could potentially impact the results. The primary focus of this work is to ascertain whether, despite these constraints, the model's performance improves when incorporating satellite data using this technique. Further investigations could delve into estimating the optimal number of rain gauges based on their distribution, representing a potential avenue for future research.

6. *L146: 1700 m3/s is a peak flow, average daily flow or what else? Some lines below the authors refer to a peak of 4350 m3/s (if "Autorità di Bacino del Fiume Po" is a reference, please add the year; if not, please explain/translate it in English).*

- A maximum flow discharge that can reach 1700 m$^3$/s in spring and autumn.
- The most significant of these events occurred in November 1994, when the entire river valley was damaged (Marchi et al., 1996; Luino, 2002) and the sensor at Montecastello, located at the outlet of the river recorded a maximum flow discharge peak of 4350 m$^3$/s (Autorità di Bacino del Fiume Po).

"Autorità di Bacino del Fiume Po" is the official Italian authority that releases information regarding the data relating to the various rivers, the document is among the references and was last viewed in November 2023. "Autorità di Bacino del Fiume Po" has been translated into English as suggested by the reviewer.

7. *L 165: Eq. (1) is quite ambiguous. It is well known that the reference area is hardly a radius, especially in orographically complex regions such as the Alps.*

This equation refers to Shi et al. Estimating a radius of influence is certainly not so immediate, but it can be a good starting point for making various considerations.

8. *L176: Earth.* Done. it has been corrected.

9. *Figure 2 and related caption: please revise. There are several errors: e.g. Guage uncal, "each case studies" [study], "eight [...] setting" [settings]. Furthermore, the terms uncal, cal, uncal1, uncal5, etc. are explained much later (L400). The explanation of the different inputs for the eight simulations should be highlighted much better (maybe with a devoted Table?).*

As suggested by the reviewer, table 2 summarizing the different inputs has been added.

10. *L325: finds.* Done. Now, LL335

11. *LL360-364: I guess this is methods, not results.*

   *Section 5.1 also is not results, but data and methods.*

   *L406: it's 5.2*

Thanks for the advice, this section has been reorganized.

12. LL418-419: however, while some of the stations considered in the study are located much further north of the watershed (as I claimed before, I believe they

are useless to this study), after the French border, there are no stations surrounding it. This drawback should be discussed.

The reviewer is right, as said before, this falls within the limits related to the selected domain, which we voluntarily wanted to maintain. In operational conditions we come across these types of problems, so the use of satellite data helps us overcome the problem, as demonstrated by the improving results.

13. LL519-528: that's methodology.

The reviewer is right, this is a repeat. the sentence has been deleted

---

## Author Response (AR2)

We thank the reviewers for the time she/he spent in the detailed reading of our work and for recognizing our efforts attempting to improve our manuscript.

Report #1

General Comments

The authors have done a reasonable job of responding to the reviews, even if pursuing approaches that I would have constructed differently.

I will note to the editor that the authors intend to use the journal's editing services to improve the English.

We thank the reviewer, We believe that after this interaction, our work has further improved.

All the changes in the revised version of the manuscript are highlighted in red color. The lines indicated for the corrections refer to the file hess-2023-214-ATC2.pdf

Specific Comments

1. L.185-186: Flow regulation is mentioned at the top of Section 5, but it should also appear here as a major challenge.

We Agree. In section 3.3 we added the following sentence:

LL183-185: *"Furthermore, data on artificial water management are not available, therefore the hydrological model validation presents difficulties in the presence of highly regulated basins since the simulation reproduces the natural flow rate of the river flow without considering the anthropogenic impact."*

We modified the LL 370-372

2. L.278: "GMP" should be "GPM".

Thanks for the correction. We did not find this error; we have checked that all the acronyms and they are correct.

3. L.409-410,463: The calibrated IMERG-F does use *monthly* gauge data, which is what makes it "calibrated". If you are thinking about *submonthly* gauge data, you need to be explicit. This addition of (monthly) gauge data is why CAL performs better than UNCAL.

We thank the reviewer for pointing this out. The rain gauge data we refer to are the local rain gauges, and not the monthly data already used in IMERG-F Calibrated. We modified the sentence.

A new reference and a new detail have been added to the LL178-180

LL454-456: *"Therefore, UNCAL and CAL simulations use only satellite data respectively IMERG-F Uncalibrated and IMERG-F Calibrated, GAUGE simulation uses rain gauge data."*

To:

*"Therefore, UNCAL and CAL simulations use only satellite data (respectively IMERG-F Uncalibrated and IMERG-F Calibrated), while the GAUGE simulation uses the local rain gauge data."*

Report #2

Although I am very well disposed toward the publication of the article, which I see as potentially valuable for the hydrological community, I am unfortunately dissatisfied with the reviewers' responses, which, on several points, seemed meagre and evasive.

As a preface, I strongly suggest that the replies to the reviewers refer clearly to the changes in the manuscript, indicating precisely the lines and sections changed.

Thanks again for the time you spent in the further revision of our work. In the following we give a precise point-by-point answer to the raised questions. We believe that after this interaction, our work has further improved.

All the changes in the revised version of the manuscript are highlighted in red color. The lines indicated for the corrections refer to the file hess-2023-214-ATC2.pdf

1. Concerning my first concern, if the authors are focused on the hydrological impact, I suggest highlighting it better both in the title (e.g., "on the combined use… to improve hydrological modelling") and the abstract.

    LL1-4: We changed the title in: "On the combined use of rain gauges and GPM satellite rainfall products for hydrological modelling: impact assessment of the cellular automata methodology on the Tanaro river basin in Italy."

    LL26-28: We modified the abstract as suggested adding the following sentence:

    "…the comparison between simulated and reference river flow discharge is crucial for assessing the effectiveness of merged precipitation data in enhancing the model's performance and its ability to realistically simulate hydrological processes."

2. More in general, I believe the authors do not consider the issue of equifinality, i.e., "the principle that in open systems a given end state can be reached by many potential means" (Wikipedia). That is related to my second concern, where I highlighted the possibility that errors can counterbalance and smooth each other. Then, much of the outcome could depend on the initial parameterization. I understand that the model was already calibrated over the whole (much larger) Po River Basin, not specifically on the Tanaro. I think this starting point weakens the paper's outcomes and should be made much more transparent.

    We thank the reviewer for this comment.

    With reference to the hydrological model, CHyM, it is worth noting that it has been widely calibrated using climatological discharge time series of the Po River, as reported in Coppola et al. (2014). To this aim, it is important to note that the conditions of the Po are representative of many alluvial rivers in Europe (Di Baldassarre et al., 2009).

    In any case, the domain used in this work which is the same one used in Coppola et al. 2014, is also the operational domain for hydrological forecasting for flood risk at CETEMPS (Centre of Excellence - University of L'Aquila). As evidenced in Figure 1 the domain does not include the entire Po basin, but only its northern part. Part of the data used in the calibration refers to a hydrometric station on the main branch of the Po River, close to the mouth of the Tanaro river, a tributary of the Po. For this reason, we believe that we can refer to that calibration, published in Coppola et al 2014, since it can be extended to the Tanaro basin without introducing significant biases.

Long simulations were carried out also for the present work to test the model configuration. Given that the purpose of the paper is different, we do not report these results.

LL342-344: This sentence is added: *"The hydrological model is not specifically calibrated over the Tarano basin. However, in this work we refer to the calibration accomplished by (Coppola at al 2014) on the northern part of the Po River, which totally includes the Tanaro basin."*

3. Concerning the third concern (description of the CA-based technique), I can't read about Coppola et al. 2010. I guess the authors refer to Coppola et al. 2007 (I also found Coppola et al. 2014, but it is not referenced).

   Thank you for pointing this out. We apologize, the reference was not correct. The correct reference is Coppola et al 2007.
    https://www.tandfonline.com/doi/epdf/10.1623/hysj.52.3.579?needAccess=true

4. In their reply, the authors claim that "the following section has been added", but I can't understand where this section is. Anyway, I still believe that some more detail should be added to make this paper more self-consistent.

   We apologize for not indicating where the sections were added: the sentence was wrong, because no section had been added in this case.
   What was done in Coppola et al 2007 is described in detail in sub-sections 4.1 and 4.2 of our paper. For this reason, the two sub-sections have been reformulated more clearly.

   in detail, in section 4.1 you will find the following changes:

[revised manuscript text omitted]

The following changes are reported in the last document uploaded in subsection 4.2:

L260-264: *"CA can be described, for example, as identical discrete sites of a lattice, and the state of each grid point evolves according to deterministic rules, conditioned by the values of neighboring cells at discrete time steps.*
*CA based algorithm has been developed and implemented in the hydrological model code. According to CA theory, the input grid is considered an aggregate of Cellular Automata, and the status of a grid point corresponds to the value of a rebuilt (i.e. smoothed) precipitation field."*

It was thus modified with the following sentence:

*"In CA, natural systems are idealized as discrete sites on a lattice, with each grid point evolving based on deterministic rules and influenced by the states of neighboring cells at discrete time steps. This approach provides a structured framework for dynamic systems modelling, reflecting the intricate interplay of elements in nature.*
*In the hydrological model code, a CA-based algorithm has been developed and implemented. Following CA theory, the input grid is conceptualized as an aggregate of Cellular Automata, where the status of each grid point represents the value of a smoothed precipitation field."*
LL279-284:*" The coefficient $\alpha$ assumes a small value (typically from 0.1 to 0.9) to ensure a slight smoothing of the original matrix: all grid points are updated synchronously, and the smoothing is*

*performed until the stability is reached, meaning that no significant changes are recorded in the calculated matrix. The grid point associated with the rainfall value available in the considered database is not modified by the algorithm. In terms of time evolution, a regular lattice is updated in discrete time steps according to the previous rule depending on the state of the site and the eight neighboring cells."*

It was thus modified with the following sentence:

*"The coefficient α assumes a small value, typically ranging from 0.1 to 0.9, ensuring a gentle smoothing of the original matrix. All grid points are updated synchronously, and the smoothing continues until stability is achieved, signifying minimal changes in the calculated matrix. Notably, the grid point associated with the rainfall value available in the considered database remains unaltered by the algorithm. This process enables the hydrological model to refine and stabilize the precipitation data while preserving the integrity of observed rainfall values."*

LL295-297:*" The CA method allows to perform the assimilation and spatialization of the rain field, it is useful for the high resolutions necessary for hydrological simulations, and to use different sources of precipitation data."*

It was thus modified with the following sentence:

*"The CA method facilitates the assimilation and spatialization of rainfall fields, proving advantageous in achieving the high resolutions required in hydrological simulations and for integrating several precipitation data sources."*

5. Concerning the rearrangement of the paper's structure, I don't understand the main changes made from this point of view, and I ask the authors to present them in the next iteration with reviewers.

In the first manuscript uploaded 01 Sep 2023 the structure was:
1 Introduction
2 Study area
3 Observed Data
    3.1 Rain Gauge data.
    3.2 Satellite-based rainfall estimates
    3.3 Observed Flow Discharge data.
4 Methodology
    4.1 Precipitation data gridding
    4.2 Precipitation data interpolation and merging: Cellular Automata technique
    4.3 Hydrological modelling: CETEMPS Hydrological Model
    4.4 Error Score Metrics
5 Analysis and discussion of the results (L359)
    5.1 Analyzed case studies. (L366)
    5.3 Results: hydrological simulation analysis (L406)
6 Conclusions (L495)

As suggested by the reviewer the structure of the paper was rearranged in the previous review.

6. Concerning the choice of using 17 scores, the authors' response is a kind of apodictic. Their very peculiar choice should be justified better. Very seldom I found that this high number of metrics was used in similar hydrological studies.

   Thanks for this remark. The reason why we chose to use 17 statistical scores is that each score highlights different, and somewhat complementary, characteristics of the simulation.

7. Specific comment no.5: in order to shed light on the issue of the number of stations needed for the Tanaro River Basin, performing some more tests by removing at least the rain gauges at the very north border of Italy could be enough. According to the authors' reply, one can conclude that also the absence of the French stations makes the work not "rigorous". Anyway, it is very odd that far-northern gauges are used while not available close western gauges are not.

   We reproduced the operational framework (domain) in which not all potential stations are effectively available. We could have selected only the river basin of interest, but we wanted to recreate the operational setting, with the same conditions and related critical issues, such as the lack of data on the French area, and to highlight that the use of satellite data is essential to overcome these critical issues.

8. Specific comment no.6: not clear yet. If the "maximum flow discharge [...] can reach 1700 m3/s, in spring and autumn", it cannot reach 4350 m3/s in November. Please rephrase.

   The reviewer is right. The 1700 m3/s of river flow discharge refers to standard seasonal conditions, while 4350 m3/s flow discharge refers to the worst condition recorded from 1801 to 2001.
   The sentence has been reformulated at LL145-147.

---

## Author Response (AR3)

We thank the reviewers for the time she/he spent in the detailed reading of our work and for recognizing our efforts attempting to improve our manuscript.

Referee #1, Report #1

General Comments

The authors have done a reasonable job of responding to the reviews.

I will again note to the editor that the authors intend to use the journal's editing services to improve the English, so I am not commenting on that aspect.

Specific Comments

1. Near-equivalence of cal and uncal combinations with gauge: Although the focus of the manuscript is on the combination method, I have been concerned that the work was done with IMERG Final, which is not available in near-real time and therefore cannot be used in a future near-real-time application. At least in this revision, it is clear that the bulk ratio adjustments used in the IMERG Final cal do not strongly influence the results compared to uncal when Final is combined with gauge in the approach being studied. This is very good news, because the near-real-time IMERG Early and Late in Version 07 have a climatological version of the gauge adjustment for cal, which cannot provide the same degree of accuracy as the month-to-month adjustment as in Final cal. But, this study seems to show that the cal/uncal distinction is not important when gauges are used. I think adding a comment to this effect when future application is discussed might be useful.

Thanks for the suggestion, we added the following sentence:

LL179-181:" Note *the IMERG Final product is considered for demonstrative purposes because such product is not available in near-real time and therefore it cannot be used in an operational context that require near-real-time constrains.*"

2. IMERG data availability: Should the specific URL for the IMERG data be stated in the data availability section? Version 05 might be hard to point to, so the editor might need to weigh in on whether the authors can just point to the current archive, which is Version 07.

Thanks for the comment: Version 05 is not immediately available for consultation, although as when there was the transition from V04 to V05 they suggested contacting them (https://gpm.nasa.gov/data-news/gpm-v05-data-announcement-and-imerg).

3. IMERG version: As an aside: for the purposes of studying the combination scheme, it is fine to use IMERG Version 5 data, but I will point out that Version 07 Final is now available for future work, and I hear Early and Late are moving to Version 07 with the start of June.

Thanks for the suggestion, we added the sentence to detail:

LL182-183: *"It is worth noting that in this study IMERG ver. 5 is used while a new version 7 is ready to be issued at the time of the publication of our study. However, the methodological value of the combination approach shown should not be affected by the product version updates, although quantitative checks could merit attention in a future work."*

Referee #2, Report #2

For final publication, the manuscript should be:
accepted as is

Thank you